# Variability in interseismic strain accumulation rate and style along the Altyn Tagh Fault

Lin Shen [1,2] ✉, Andrew Hooper [1], John R. Elliott [1] & Tim J. Wright [1]

Major strike-slip faults that develop between strong and weaker regions are thought to focus along narrow shear zones at the rheological boundary. Here we present the InSAR-derived velocity field spanning almost the entire length of one such fault, the 1600 km-long Altyn Tagh Fault (ATF), and analyse the strain distribution. We find that localisation of strain is actually variable, in contrast to other major strike-slip faults that show little variation, with strain concentrated at the fault for some sections and distributed over broad (>100 km) shear zones for others. Slip rate along the ATF is also variable, decreasing along the fault from 11.6 ± 1.6 mm/yr in the west to 7.2 ± 1.4 mm/yr in the central portion, before increasing again to 11.7 ± 0.9 mm/yr over the eastern portion. We show that the variable shear zone width may be linked to geological variability and the influence of heat flow, and the results imply that sub-parallel faults play an important role in the overall deformation field. This demonstrates the significance of accurately characterising strain rates over a broad region when assessing seismic hazard.

Major intra-continental strike-slip faults that have developed adjacent to strong regions are usually considered to absorb large fractions of relative plate movement between lithospheric plates and deforming regions, marking discontinuities in the long-term velocity field[1]. The North Anatolian fault in Türkiye, near the relatively strong oceanic lithosphere of the Black Sea, is such a case, where strain rate across the fault varies little along the entire length[2]. Similarly, the strain rate along the San Andreas fault system in California is concentrated along a single trace where it goes around the Central Valley[3].

The 1600 km-long Altyn Tagh Fault (ATF) is a major intra-continental strike-slip fault in the Northern Tibetan Plateau (Fig. 1). Tectonic processes in this region are not fully understood, and the debate about how continental deformation accommodates the Indo-Asian plate collision still continues[4]. According to the United States Geological Survey (USGS) earthquake records, a pair of earthquakes occurred along the western portion of the ATF in 1924, with magnitudes of $M_w$ 7.0 and $M_w$ 7.2, respectively. In 1932, a $M_w$ 7.6 earthquake occurred along the Changma fault, near the easternmost part of the

ATF. Following the 1932 earthquake, no major earthquake ($M_w > 7.0$) has been recorded along the ATF[5].

Bounding to the rigid Tarim Basin, the ATF was thought to have localised strain in a narrow zone in previous studies [e.g., refs. 6,7], marking discontinuities in strength moving from the relatively strong Tarim Basin in the north to the deforming Tibetan Plateau to the south. However, in contrast to previous studies on other major intra-continental strike-slip faults, such as the North Anatolian Fault, which shows a focussed strain along its entire length [e.g., refs. 2,8], it is unclear how much of the total strain across the ATF is concentrated at the fault itself and how much is more broadly distributed. Recent slip rate estimates along the ATF are generally determined by three methods over different time scales: (i) long-term geological measurements through the identification of offset (displaced) piercing points across the fault [e.g., ref. 9]; (ii) Quaternary estimates based on stream, terrace offsets, cosmogenic or [14]C dated offsets [e.g., ref. 10]; (iii) Geodetic modelling using Global Navigation Satellite System (GNSS) and Interferometric Synthetic Aperture Radar (InSAR) measurements [e.g., refs. 11–13]. Although these studies have provided valuable

---

[1]COMET, School of Earth and Environment, University of Leeds, Woodhouse Lane, Leeds LS2 9JT, United Kingdom. [2]Lamont-Doherty Earth Observatory, Columbia University, 61 Rte 9W, Palisades, NY 10964, USA. ✉e-mail: lshen@ldeo.columbia.edu

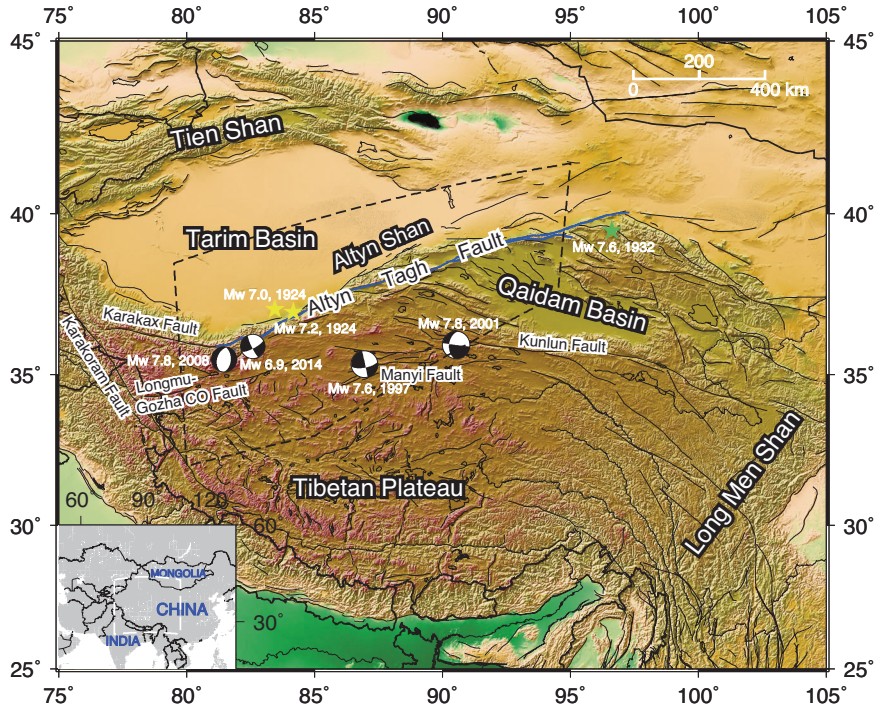

**Fig. 1 | Tectonic setting of the Altyn Tagh Fault (ATF) zone and other major features in Tibet.** The black dashed lines indicate the extent of the study region. Black solid lines represent fault traces in the region[65]. Blue lines highlight the high-resolution fault traces of the main strand of the ATF[66], which we use in the later interseismic deformation modelling. Yellow stars represent a pair of earthquakes that occurred along the western portion of the ATF in 1924, with magnitudes of $M_w$ 7.0 and $M_w$ 7.2, respectively. Major earthquakes ($M_w > 6.9$) that occurred near the ATF recently are also featured in the map, including the 1932 $M_w$ 7.6 Changma earthquake (shown as green star), 1997 $M_w$ 7.6 Manyi earthquake, the 2001 $M_w$ 7.8 Kokoxili earthquake, the 2008 $M_w$ 7.2 Yutian earthquake, and the 2014 $M_w$ 6.9 Yutian earthquake. The map background shows the elevation of the study region derived from the Shuttle Radar Topography Mission (SRTM) 3-arc seconds data[67].

insights into the interseismic deformation along the ATF, they have only focussed on specific portions and do not provide an overall picture of the variation of localised strain accumulation along the fault.

Additionally, published geodetic measurements along the ATF suggest 0–10 mm/year of slip rate over the western portion from 78 °E to 80 °E [e.g., refs. 14,15], 5–15 mm/year for the central portion from 84 °E to 90 °E [e.g., refs. 16–19] and 4–10 mm/year over the eastern portion [e.g., ref. 15,20], which are not always in agreement with those derived from long-term geological measurements nor Quaternary rates. For instance, slip rates inferred from geodetic data are generally a factor of 2 to 3 lower than those inferred from geologic data [e.g., ref. 21]. The discrepancy could be caused by the uncertainties of the measurements[22], or may indicate a secular change in fault slip rates over distinct time scales[23]. Mohadjer et al.[22] applied least squares regression to the pairs of published GNSS and geological slip rates along the ATF and suggested that disagreements between the GNSS and Quaternary rates can mainly be ascribed to the incorrect geomorphic reconstructions of offset landforms used for estimating Quaternary slip rates. Thatcher[24] also suggested that those high geological estimates are controversial. As very small differences in the data can translate into significant intrablock deformation and slip on the fault, the precision of slip rate constraints for the ATF is crucial to explain how continents deform there.

Here we present a new InSAR-derived interseismic velocity field spanning almost the entire length of the ATF from Sentinel-1 interferograms over five years, between late 2014 and 2019, which is the first time a large-scale analysis covering almost the entire length of the fault has been carried out with a resolution sufficient to identify areas of strain localisation. To derive a consistent velocity field over an extensive length scale, we develop a novel method to stitch InSAR velocity fields estimated from different satellite tracks. We apply two distinct models, a modified deep-fault model, and a distributed shear

zone model, to investigate the variability in both the rate and style of interseismic strain accumulation along the ATF. We also assess the earthquake potential in the region by consolidating the insights obtained from the modelling.

## Results

### Large-scale InSAR-derived velocity field

We estimate an InSAR-derived velocity field over ~1500 km of the ATF, between 80 °E to 95 °E (Supplementary Figs. 1 and 2, Equation (1)). To derive a consistent velocity field over the extensive spatial scale, we develop a new method for mosaicking InSAR line-of-sight (LOS) velocities over adjacent satellite tracks with a diverse range of viewing geometry (Equation (2), (3)). We resample the merged InSAR LOS velocities onto a 1 km by 1 km grid using a nearest neighbour resampling algorithm. We compare the velocities before and after resampling along short fault-parallel velocity profiles with a length of 10 km on either side of the ATF. To avoid any further resampling in the comparison, we show the velocity of points located within a spatial distance of 5 metres before and after resampling, individually. The results indicate a negligible difference in both ascending and descending directions (Supplementary Fig. 3). We then decompose the velocity field into an east-west and a sub-vertical component, which is mostly sensitive to vertical movement but is also somewhat sensitive to north-south movement (Equation (2)). This approach enables us to resolve the east-west velocities unambiguously. The derived east-west velocity map shows a clear gradient across the ATF resulting from the eastward motion of the Tibetan Plateau with respect to the Tarim Basin (Fig. 2a). Over the south-western strand of the ATF, west of 83 °E, the map shows that the strain accumulation transfers to the structurally linked ENE-striking left-lateral strike-slip Longmu-Gozha Co Fault (LGCF), through the Ashikule step-over zone. Regions of uplift or northward motion observed to the north of ATF from 87 °E to 89 °E in

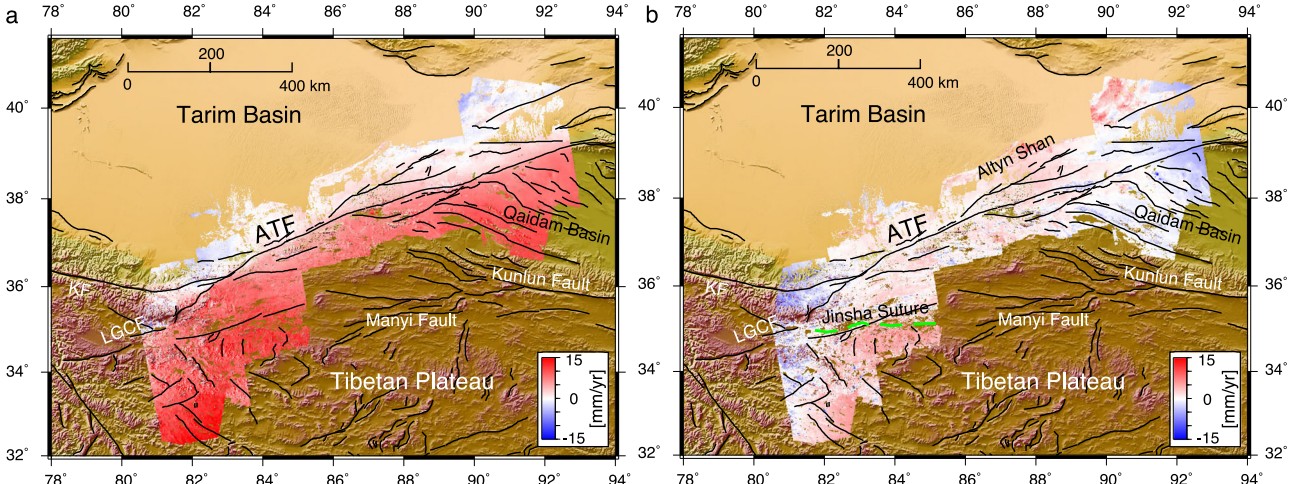

**Fig. 2 | InSAR-derived velocity maps in the Tarim reference. a** East-west velocity map. Positive values indicate the eastward motion. **b** Sub-vertical velocity map. Positive values indicate the northward or uplift motion. The green dashed line shows the trace of the Jinsha suture derived from Daout et al.[25]. ATF Altyn Tagh Fault. KF Karakax Fault. LGCF Longmu-Gozha Co Fault. The map background shows the elevation of the study region derived from the Shuttle Radar Topography Mission (SRTM) 3-arc seconds data[67].

the sub-vertical component (Fig. 2b), can potentially be ascribed to the south dipping Altyn Shan thrusts along the southern border of the Tarim Basin[25]. In the region north of the ATF from 81.5 °E to 82.5 °E, the observed sub-vertical motion could be associated with southward movement resulting from left-lateral strike-slip of the fault. The sub-vertical component also shows a signal over the eastern edge of the Tarim Basin that does not appear tectonic. We interpret this as being associated with the hydrological processes or effects of the sand dunes there.

**Variable slip rates along the ATF**

Considering the variable strike along the ATF, we derive 1-D fault-parallel velocity profiles at intervals of 0.5 °, directly decomposed from the InSAR LOS velocities, based on a varying local strike perpendicular to the high-resolution fault trace. The derived fault-parallel velocity profiles, projected using data within a 50 km perpendicular distance, show visible strain accumulation on the ATF (Supplementary Fig. 4). The asymmetric pattern of interseismic velocities shown in the profiles suggests a decrease in rigidity from the Tarim Basin to the Tibetan Plateau[12,26,27]. The profiles also reveal that additional strain localisations are distributed over southern strands near the western portion of the ATF from 84 °E to 85.5 °E and the eastern portion from 91 °E to 92 °E. We also project previous modelling of GNSS measurements[15,28] into the fault-perpendicular profiles, and the results show that they are in good agreement (Supplementary Fig. 4).

We derive the slip rate and locking depth from the fault-parallel velocity profiles using a modified deep-fault model (Equation (4)). To explain the asymmetry in the interseismic velocities on each side of the fault, we solve for a rigidity ratio as an asymmetry coefficient in the model to characterise the differing rigidity between the Plateau and the Tarim Basin. As previous studies show the proximity to the Euler pole of the rotation can lead to additional velocity variation in the fault-parallel velocity [e.g., ref. 29], we estimate a rotation rate for each profile south of the fault in the model to account for this. We fit a linear trend through the resolved fault-parallel far-field GNSS velocities south of the fault and the estimated mean rotation rate is 0.0122 mm/yr/km. Therefore, we constrain the rotation rate between 0 and 0.03 mm/yr/km in the modelling (Equation (4)). We also solve for additional slip rate, locking depth and buried dislocation shift on secondary faults (Equation (5)) for the profiles from 84 °E to 85.5 °E and from 91 °E to 92 °E.

The interseismic modelling results reveal a systemic apparent decrease in fault slip rate along the ATF (Fig. 3b, Supplementary Fig. 5),

from 11.6 ± 1.6 mm/yr along the western portion (from 80 °E to 84 °E) to 7.2 ± 1.4 mm/yr along the central portion (from 84 °E to 88 °E) of the fault, whereas it increases to 11.7 ± 0.9 mm/yr over the eastern portion (from 88 °E to 93 °E).

The posterior probability distributions (Supplementary Fig. 6) for the locking depth are generally less than 20 km when the fault is shown as a single strand from 80 °E to 87.5 °E (Fig. 3c), which supports the small thickness of the seismogenic layer in the lithosphere (not exceeding ~20 km) suggested by previous depth distribution of earthquakes in this region[30]. We interpret the higher estimates of locking depth from 87.5 °E eastward as a wider zone in which the interseismic strain is accommodated due to splitting of the fault into three strands. The estimated buried dislocation for the main strand of the ATF is located south of the fault from 81.5 °E to 83 °E. It shifts slightly to the north of the fault along the central segment, then interweaves with the fault trace over the eastern portion. Additional buried dislocations are required for the profiles from 84 °E to 85.5 °E, located ~50 km to 150 km south of the ATF (Fig. 3a), where other sinistral faults are mapped[31]. For the profiles over the eastern portion from 91 °E to 92 °E, the buried dislocation shifts ~120 km southward to a region that features other multiple fault strands. The multimodal marginals shown in the posterior probability distributions suggest that the strain could be distributed along multiple strands of the ATF (e.g., at 82 °E, 90 °E, 90.5 °E and 91 °E) or may be influenced by relatively noisy data (e.g., at 89.5 °E). The modelling results indicate that no significant creep occurs along most portions of the ATF, except for higher creep rates of over 2 mm/yr on the westernmost end of the fault and the region from 89 °E to 89.5 °E (Fig. 3d), at depths shallower than 20 km (Fig. 3e). Although shallow creep is not observed in the modelling results from the GNSS measurements between 88 °E and 91 °E[18], this could be due to the use of a wider profile (~300 km), which may be too smoothed to detect small-scale shallow creep. The estimated creep rate is not highly correlated with the estimated locking depth, as the locking depth of areas with higher estimates of creep rate varies from 7 km to 40 km (Supplementary Fig. 7), demonstrating that the model adequately captures the strain required to fit the observation. As expected, we observe a reduction in rigidity from the Tarim Basin to the Tibetan Plateau, with an average rigidity ratio of 0.60 (Supplementary Fig. 8). This agrees with the understanding that the Tarim Basin was relatively stable during the Cenozoic[32]. Moreover, we identify a higher rigidity ratio of up to 0.99 in the west-central segment from 83 °E to 86 °E, indicating a significant contrast in rigidity across this region.

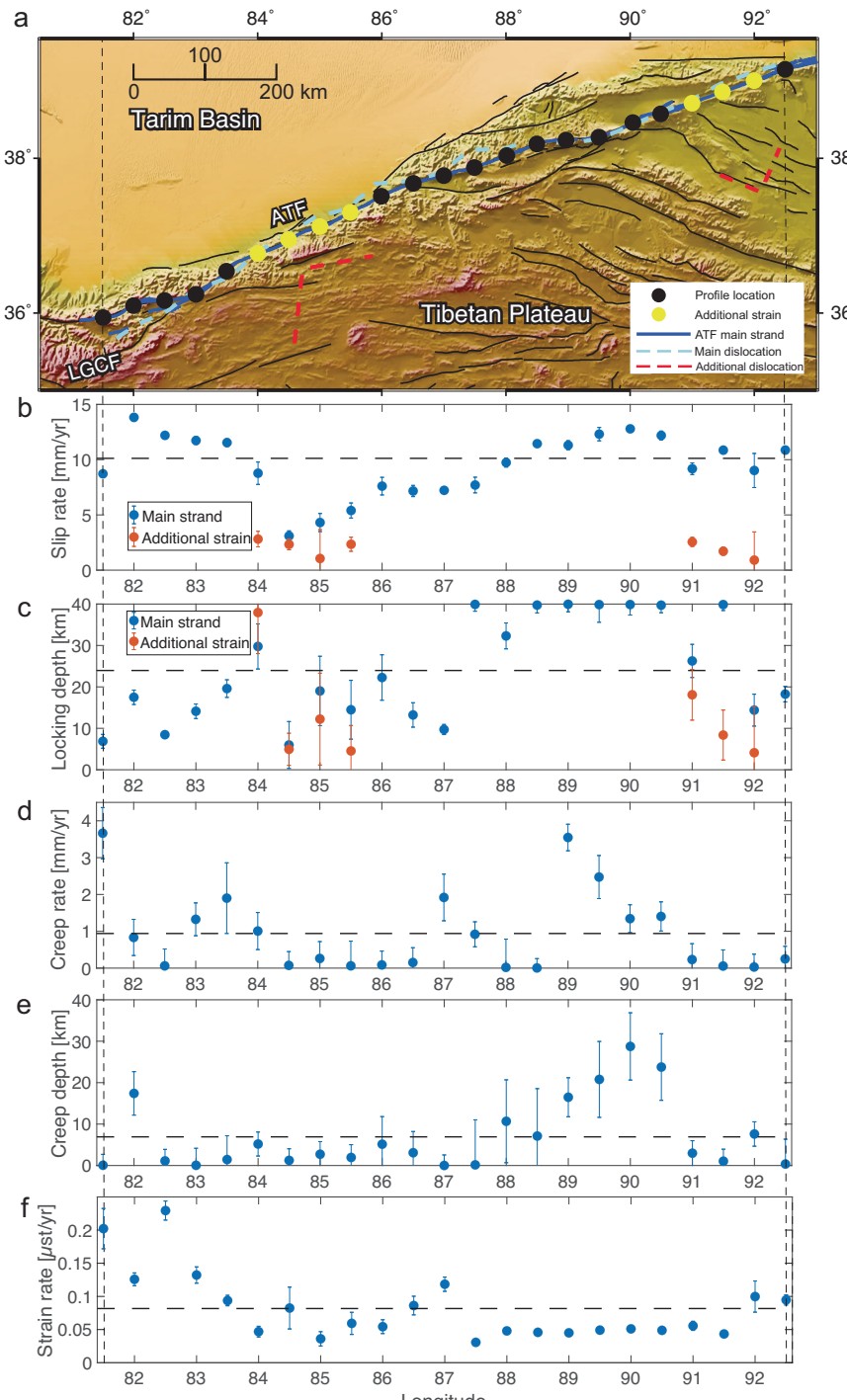

**Fig. 3 | Estimates of slip rate and strain rate along the Altyn Tagh Fault (ATF) based on a modified deep-fault model. a** Location map of estimated buried dislocations. The light blue dashed line indicates the estimated locations of the buried dislocations along the ATF, and the red dashed lines show the estimated locations of the buried dislocations away from the ATF. Blue bold lines represent the high-resolution fault traces of the main strand of the ATF[66]. The estimated additional buried dislocations for the profiles from 84 °E to 85.5 °E are located where other sinistral faults are mapped[31]. For the profiles over the eastern portion from 91 °E to 92 °E, the buried dislocation shifts southward to a region that features other multiple fault strands. LGCF = Longmu-Gozha Co Fault. The map background shows the elevation of the study region derived from the Shuttle Radar Topography Mission (SRTM) 3-arc seconds data[67]. **b–f** Parameter estimates along the ATF based on the modified deep-fault model. The black dashed lines give the average estimate for each parameter. The error bars represent the 68% confidence bounds on the parameter estimates. $\mu$st = $\mu$strain.

## High strain localisation over the south-west portion

Based on the strain rate estimation (Equation (6)), we find a consistent peak strain rate at the surface along most of the fault (Fig. 3f), with an average value of 0.08 $\mu$strain/yr. However, we find a high strain rate greater than 0.2 $\mu$strain/yr is accumulating at the surface along the south-western strand of the ATF, a region that is hardly covered by previous studies. Our results show that the clear strain accumulation is structurally linked to the LGCF in the west through the Ashikule step-over zone, rather than transferring to the Karakax Fault. This step-over region has been highly active recently: four major earthquakes ($M_w$ >

6.3) have occurred in 2008, 2012, 2014 and 2020, including the 2008 $M_w$ 7.2 Yutian normal faulting earthquake[33], which is the largest normal faulting event ever recorded in northern Tibet, the 2012 $M_w$ 6.3 normal faulting earthquake and the 2014 $M_w$ 6.9 Yutian strike-slip earthquake. The estimated slip rate of ~12 mm/yr along the south-western segment of the ATF is much higher than previous estimates of the rate of the LGCF fault, which were on the order of 4 mm/yr, although made further south-west[14,34].

Our results suggest that the generation of the SN-trending normal faulting events in the region is ascribed to the EW-trending extensional stress at a step-over between the two left-lateral faults[35], where the North-South shortening occurs. These large earthquakes have significantly increased the stress over the region. Bie et al.[36] calculated the combined stress loading effect of the 2008, 2012 and 2014 earthquakes on the ATF and found that both the 2008 and 2012 normal faulting earthquake exert positive Coulomb stress changes to the 2014 strike-slip earthquake rupture. Li et al.[37] found that the 2014 Yutian strike-slip earthquake has further increased the Coulomb failure stress on the south-western segment of the ATF. Therefore, the high strain rate estimated over the south-western portion might be ascribed to the stress loading effects of the recent seismic activities.

To investigate the impact of postseismic deformation following the February 2014 $M_w$ 6.9 Yutian strike-slip earthquake, which occurred nine months before the InSAR observations of this study, we calculate the time series of relative LOS displacement between two sites located around 30 km apart, either side of the south-western segment(Supplementary Fig. 9). We calculate the average LOS displacement of points within 2 km distance to each site from the tropospheric-corrected single primary interferograms, to form a time series of relative LOS displacement between the two sites, to which we fit a linear trend. We find a generally consistent rate in the ascending track. For the descending track, the displacement in the early time series (673 days before the earthquake occurred) has a systematic bias, indicating a higher rate before November 2015 compared to the later time series. We also fit a logarithmic decay trend (Equation (7)) to the time series. While other equations are available to model time-dependent deformation [e.g., ref. 38], we find that the simple one we choose performs well. Compared to the linear model, there is no significant improvement when fitting the data with the logarithmic model. This suggests that the postseismic deformation is hard to distinguish from the long-term interseismic deformation. Consequently, while the presence of the high strain rate on the south-western segment of the ATF raises the possibility of a relatively greater seismic risk in this region, it remains challenging to assess the impact of the postseismic deformation following the 2014 Yutian earthquake.

### Along-strike changes in shear zone width

As the results from the modified deep-fault model suggest that the strain along some portions of the ATF is likely widely distributed, we fit the fault-parallel velocity profiles with a distributed shear zone model (Equation (8)) to investigate the distributed shear strain in a broader shear zone. We fix the locking depth at 15 km to allow for the modelling of broadly distributed deformation across the shear zone. The modelling results reveal three sections with a broad shear zone along the ATF: ~127 km between 83.5 °E to 85.5 °E, ~110 km between 87.5 °E to 88.5 °E and ~136 km between 90 °E to 91.5 °E (Fig. 4a, b). In each of these sections, we find localised strain primarily on the main strand of the ATF but also distributed within a wide region. Each broad shear zone intersects with a series of other mapped sinistral faults and significant tectonic features, such as the Altyn Shan in the central broad shear zone and the Qaidam Basin in the eastern broad shear zone (Fig. 4c). Additionally, the estimated buried dislocation in each broad shear zone region corresponds to the presence of nearby sub-parallel faults or other mapped geological structures. The two broad shear zones identified between 83.5 °E to 85.5 °E and between 90 °E to 91.5 °E

align with the region where we infer additional strain localisations from the fault-parallel velocity profiles. According to the USGS earthquake catalogue, strands in these areas have been active recently, recording four earthquakes ($M_w$ > 5.0) in the year of 1960, 2000, 2007 and 2016, respectively (Fig. 4a). The broad shear zone between 87.5 °E to 88.5 °E is situated where the ATF breaks into three parallel strands. The wider shear zones explain the high estimates for the locking depth solved from the modified deep-fault model in these areas. However, the estimated shear zone width is ~0.2 km at 89 °E and ~3.1 km at 89.5 °E, which is much narrower than the two neighbouring sections with a broader shear zone width. The narrow shear zone width in this area is likely a result of the relatively noisy data, which cannot distinguish between a possible shallow creep using the deep-fault model or a localised single fault using the distributed shear zone model.

Based on the fault-parallel velocity profiles, smoothed by a median filter with a window size of 80 km, we estimate the strain rate for each profile and find multi-peak strain rates over the three broad shear zones (Fig. 4b). The results suggest that the strain over the wide shear zones could be distributed along multiple strands of the ATF or across other fault strands that are away from the ATF, where similar split strands can be observed in other fault systems such as the northern Alpine Fault[39] and the southern San Andreas Fault[40]. In most cases, the main strand of the ATF coincides with the highest peak in strain rate.

To quantify the continuation of the main strand of the ATF, we estimate slip rates for short fault-parallel velocity profiles with a length of 50 km on either side of the main strand, using the modified deep-fault model, and compare with the overall deformation estimated from the distributed shear zone model. While the ATF bounds the rigid Tarim Basin, we find that the localisation of strain varies along its length (Fig. 5). Our results indicate that strain is distributed over 53% of the fault's length, which is larger than the proportion of the length where strain is concentrated at fault. This suggests that a significant portion of the relative plate movement is absorbed by sub-parallel faults near the ATF, indicating their important role in the overall deformation field.

## Discussion

Given the similarity in driving forces along the fault system, geological heterogeneity could potentially explain the variability in shear zone width along the ATF. Rheological properties of rocks, influenced by parameters such as composition, temperature, water fugacity, and melt content, play a significant role in determining the width of shear zones[41]. It has been observed that shear zone width is inversely proportional to rock strength, with temperature being a key factor [e.g., ref. 42]. Moreover, the specific combinations of minerals with suitable properties can result in intense localisation within shear zones[43]. Based on a geological map from the USGS[44] showing the distribution of the geologic age and type of surface outcrops of bedrock in the Northern Tibetan Plateau (Supplementary Fig. 10), the estimated shear zone width along the ATF decreases in the presence of intrusive rocks. This is likely due to the higher strength and rigidity of intrusive rocks compared to surrounding rocks, creating barriers to the movement of the fault, and leading to localised deformation.

Furthermore, the reactivation of suture zones has been shown to be an important process that controls the location of active faults such as the North Anatolian Fault [e.g., ref. 45]. Tibet was constructed from a series of distinct geological terranes and the sutures between them have been hypothesised to act as faults in some locations[46]. The ATF intersects different terranes as well as distinctive geological structures such as sedimentary basins[47]. These variations in geology are likely to lead to rheological variations that can influence fault localisation and propagation, resulting in variable shear zone width. Specifically, the broad shear zone width from 87.5 °E to 88.5 °E can be attributed to the interaction between the ATF and the Altyn Shan. The tectonic activity associated with the Altyn Shan, such as faulting and folding, can lead to

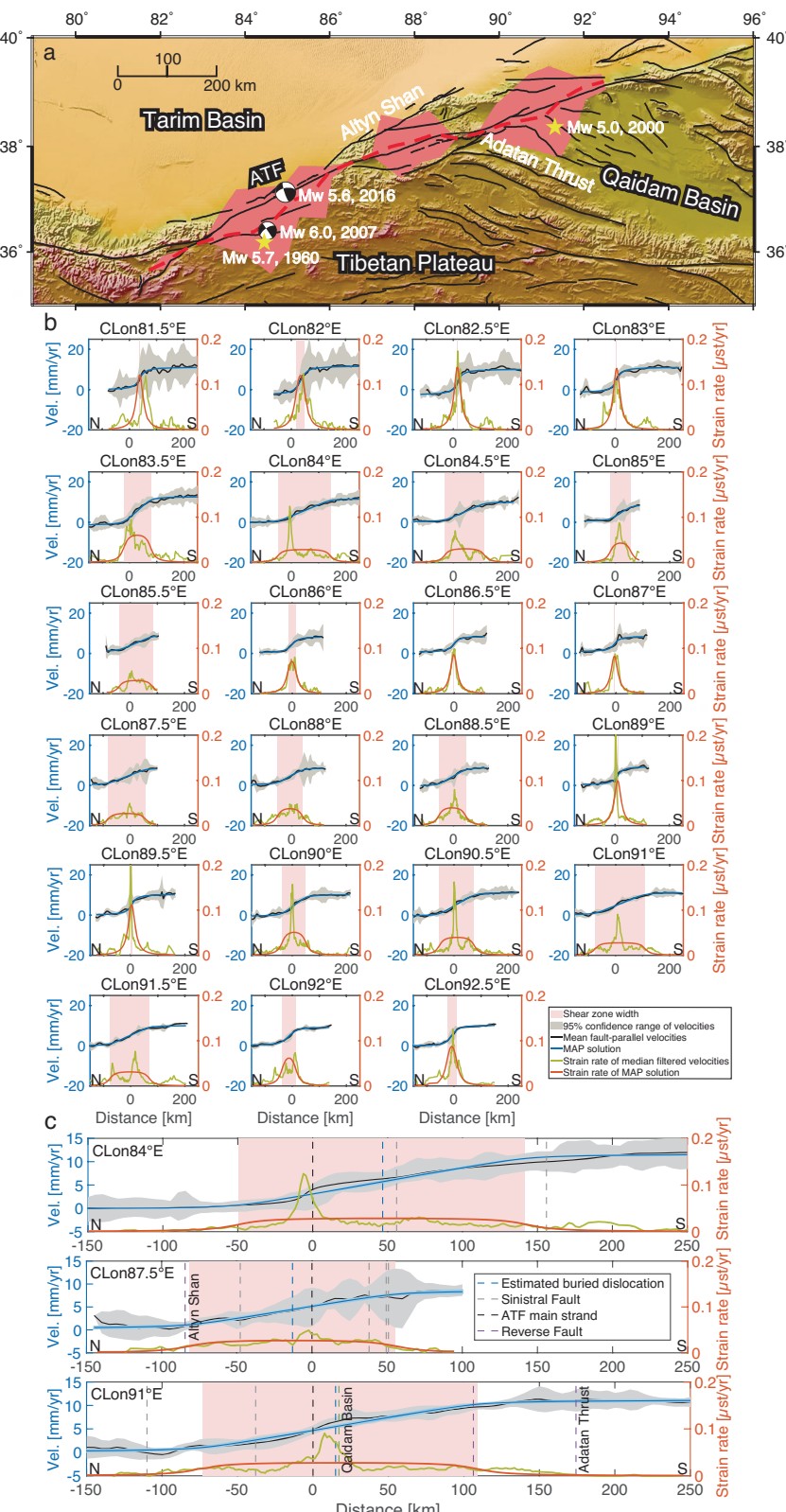

fracturing along the mountain range[48]. This creates zones of weakness that are more susceptible to deformation, contributing to a wider shear zone. Moreover, the topographic relief of the Altyn Shan may result in the redistribution of stress in the crust and influence the behaviour of nearby faults, including the ATF. The region with a wide shear zone width from 90 °E to 91.5 °E is located where the area intersects with the Qaidam Basin. The Qaidam Basin is characterised by

thick sedimentary sequences[49], which typically have lower strength and higher ductility, allowing them to accommodate strain over a wider area.

Regarding the geothermal characteristics, high heat flow can increase the temperature in the crust, reducing the strength of rocks and making them more prone to deformation. Additionally, higher heat flow can drive fluid migration within the crust, facilitating fault

**Fig. 4 | Estimated shear zones along the Altyn Tagh Fault (ATF) based on a distributed shear zone model. a** Location map of estimated shear zones. Light red areas represent the locations of the estimated shear zones along the ATF, and red dashed lines indicate the locations of the buried dislocations of the ATF estimated from the distributed shear zone model. Yellow stars and focal mechanism solutions represent four earthquakes ($M_w > 5.0$) which have occurred recently in the region that features broad shear zones. The map background shows the elevation of the study region derived from the Shuttle Radar Topography Mission (SRTM) 3-arc seconds data[67]. **b** Estimates of shear zone width and strain rate based on the distributed shear zone model. The light grey-shaded region represents the 95% data confidence range of each fault-parallel velocity profile, and the black bold line represents the mean velocities binned by every 5 km along the profile. Blue curves show the maximum a posteriori (MAP) solution of the distributed shear zone model for each profile. Yellow-green and orange curves represent the strain rate estimated from the profile median filtered with a window size of 80 km and the MAP solution, respectively. Vel. = Velocity. $\mu$st = $\mu$strain. **c** Three enlarged profiles for each broad shear zone, located at 84 °E, 87.5 °E, and 91 °E, respectively. The light blue-shaded region represents the 95% model confidence range.

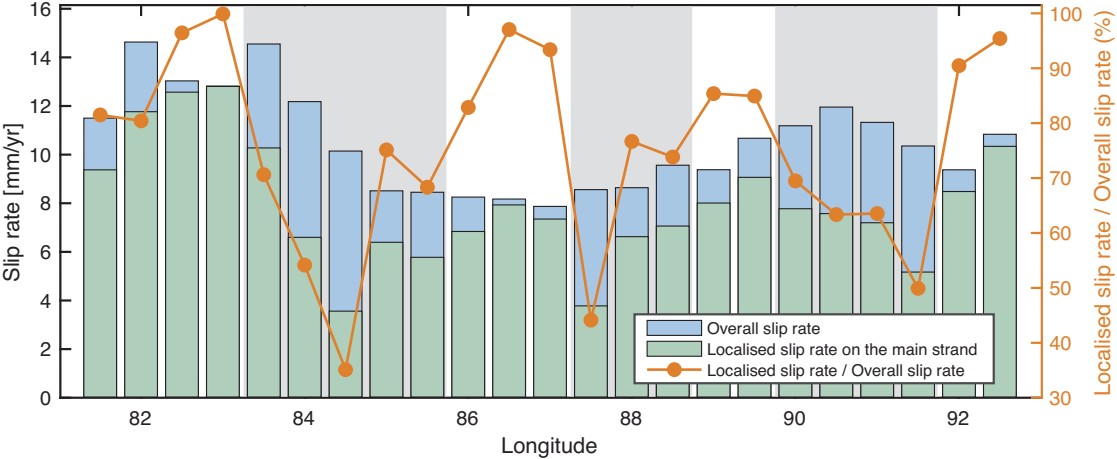

**Fig. 5 | Comparison of localised slip rate on the main strand and overall deformation along the Altyn Tagh Fault.** Light grey areas indicate the presence of three broad shear zones.

movement and potentially widening the shear zone[50]. The heat flow distribution map in continental China shows relatively higher heat flow coinciding with the wide shear zone regions along the ATF from 83.5 °E to 85.5 °E and from 90 °E to 91.5 °E[51]. Furthermore, the western Qaidam Basin has relatively high heat flow[52], which can further facilitate distributed deformation along a broader zone.

In addition to geological and geothermal variations, past earthquake events can redistribute stress and strain in the crust, impacting fault and shear zone behaviour. Previous studies indicate that recent large earthquakes in the south-western portion of the ATF have significantly changed stress in this region [e.g., ref. 37], potentially contributing to the variable shear zone width observed in the south-west of the fault.

In this study, we present an InSAR-derived velocity field along 1500 km of the ATF, spanning almost the entire length of the fault. The interseismic modelling results suggest a systemic decrease of the slip rate along the ATF from $11.6 \pm 1.6$ mm/yr to $7.2 \pm 1.4$ mm/yr over the western portion to the central portion, whereas it increases again to $11.7 \pm 0.9$ mm/yr over the eastern portion. Along the south-western segment of the ATF, we find that the strain accumulation transfers to the structurally linked Longmu-Gozha Co Fault in the west through the Ashikule step-over zone, with a high strain rate greater than 0.2 $\mu$strain/yr. This demonstrates that the generation of the NS-trending normal faulting events over the region is ascribed to the EW-trending extensional stress at a step-over between the two left-lateral faults and an east-south-eastward propagation of the faulting into the plateau, rather than more northwards around the western edge of the Tarim Basin. We find three discrete broad shear zones wider than 100 km along the ATF, where the strain distributes across multiple strands of the ATF itself or other fault strands that are away from the ATF. The along-strike changes in shear zone width are likely attributed to local geological heterogeneity and geothermal variation. Our findings distinguish the ATF from other major strike-slip faults that show little variation in the localisation of strain, suggesting the importance of considering the impact of sub-parallel faults in understanding tectonic processes over the Northern Tibetan Plateau.

## Methods
### InSAR data and processing
We process 12 overlapping tracks (6 ascending and 6 descending) using the first 5 years of Sentinel-1 SAR data, spanning the period between late 2014 and 2019 (Supplementary Fig. 1). The original InSAR data includes 21 frames (250 km by 250 km) defined by the Sentinel-1 processing system LiCSAR[53] and we process the equivalent of 1–2 frames along each track (250–500 km-long). To avoid possible inconsistent phases in the overlapping region of the same track during the construction, we merge frames within the same track to form a long track before generating interferograms using LiCSAR (Supplementary Fig. 11).

We generate short temporal baseline interferograms forming a small baseline network with sufficient redundancy, which has 200 interferometric pairs on average (Supplementary Fig. 12). We use the Stanford Method for Persistent Scatterers (StaMPS) software[54] to select highly coherent slowly-decorrelating filter phase scatterers from the interferograms, before downsampling and unwrapping the interferometric phase of the stable scatterers. As the ATF is located at the border between the lower Tarim Basin and the high Tibetan Plateau, the expected long-wavelength deformation signal is strongly masked by tropospheric delay variation across the 6 km of topographic relief. To improve the retrieval of small tectonic signals, we apply a spatially varying scaling method based on weather models[55] to reduce the tropospheric effects. After removing the estimated tropospheric delays from the interferograms, the InSAR phase no longer has strong correlations with the topography, implying significant reductions in tropospheric signals (Supplementary Fig. 13a–l). The average root mean square for all tracks drops by 37% after corrections (Supplementary Fig. 13m). Compared to descending tracks, the improvement on the

ascending tracks, which have greater noise, is more significant after the correction is applied.

We use the short temporal baseline interferograms to invert for interferograms with respect to a single primary image. As the accumulated displacement is an aggregation of short temporal data, it provides both short-term and long-term signals (Supplementary Fig. 14). We then derive the InSAR LOS velocity fields for each pixel from the tropospheric-corrected interferograms using the function

$$D_{\mathrm{LOS}} = a \sin\left(\frac{2\pi}{365.25}\right)t + b \cos\left(\frac{2\pi}{365.25}\right)t + vt + c \qquad (1)$$

where $D_{\mathrm{LOS}}$ is the InSAR accumulated LOS displacement, the sinusoidal term $a\sin(2\pi/365.25)t$ and the cosinusoidal $b\cos(2\pi/365.25)t$ term model signal subject to the annual seasonal freezing and thawing of the upper layer of the permafrost in Tibetan Plateau, $v$ is the annual LOS velocity, $t$ is the time from the primary date of interferograms of each track and $c$ is a constant. While we are not interested in the seasonal signal in itself, including it in the estimation allows for a more accurate estimation of the velocity. We invert the solution using the best linear unbiased estimator (BLUE) [e.g., ref. 56]. We calculate phase variances for each epoch from the variances of the tropospheric-corrected short temporal interferograms by least squares inversion. We then use these variances as the elements on the principal diagonal of the variance-covariance matrix in the BLUE inversion. Off-diagonal elements are set to zero since the noise of each epoch is treated as independent. We calculate standard deviations of the velocity for each pixel using the percentile bootstrap method [e.g., ref. 57], which indicates the threshold for measuring tectonic signals.

## InSAR velocity mosaicking over adjacent tracks

Variation in satellite geometry (e.g., azimuth direction and incidence angle) and long wavelength errors between tracks could lead to velocity inconsistencies in the overlapping regions, which is non-negligible in large-scale studies. To obtain a consistent velocity field across the region, we developed a new method, first presented by Shen[58], to estimate long wavelength trends from the InSAR velocity fields using GNSS observations within the region[15,28] and meanwhile minimising the differences between the adjacent tracks of the InSAR LOS velocity fields. We remove the long wavelength trends from the InSAR LOS velocity fields and decompose the velocities into an east-west component and a sub-vertical component, which we define below.

We incorporate three kinds of InSAR points in the inversion for long wavelength trends: (i) points within the overlapping regions across tracks that have both ascending and descending measurements, to minimise the velocities differences between adjacent tracks; (ii) points that have both ascending and descending measurements and are also co-located (within a 1 km distance) with GNSS observations of horizontal (only) velocities; (iii) points which are co-located (within a 1 km distance) with GNSS observations of both the horizontal and vertical components. Constraints from the latter two kinds of points reference the InSAR velocity field to the GNSS reference frame. Each horizontal GNSS point can provide one extra constraint in the inversion, and each GNSS point with the vertical component can provide two extra constraints.

To remove the velocities resulting from continental rotation, we first transform the GNSS horizontal velocities from a Eurasia-fixed reference frame to a Tarim reference, by minimising the velocities of the GNSS stations located in the Tarim Basin (Supplementary Fig. 15). The GNSS horizontal velocities referenced to the Tarim Basin indicate clear left-lateral strike-slip motion along the ATF. As we only have two observational InSAR inputs in the form of ascending and descending, it is not possible to incorporate the full 3-D velocity field in the inversion. Previous studies along the ATF from InSAR measurements often ignore the north-south component due to the lack of evidence of fault shortening over this region [e.g., ref. 17]. Alternatively, some studies use the interpolated north-south GNSS velocities as the constraint [e.g., ref. 8], although the accuracy is highly dependent on the density and spatial distribution of the GNSS sites. In this study, we consider an east-west striking plane, tilted to the south, representing the average plane defined by all LOS vectors. We estimate two components of the crustal velocity in this plane: the east-west direction and the "sub-vertical" direction, perpendicular to the east-west direction as

$$V_{\mathrm{LOS}} = \left[ -\sin\theta\cos\phi, \quad \sqrt{((\sin\theta\sin\phi)^2 + (\cos\theta)^2)} \right] \begin{bmatrix} V_e \\ V_{\mathrm{subv}} \end{bmatrix}$$
$$\text{where} \quad V_{\mathrm{subv}} = \left(\frac{\sin\theta\sin\phi}{\Delta}\right)V_n + \left(\frac{\cos\theta}{\Delta}\right)V_u \qquad (2)$$
$$\Delta = \sqrt{((\sin\theta\sin\phi)^2 + (\cos\theta)^2)}$$

where $V_e$ is the east-west component, $V_{\mathrm{subv}}$ is the sub-vertical component, $V_n$ and $V_u$ are the northern and vertical motion, respectively, $\theta$ is the radar incidence angle, and $\phi$ is the azimuth direction of the satellite positive clockwise from the north.

By inverting all the GNSS and sampled InSAR velocities, we can estimate the long wavelength trends in the InSAR velocities for each track by solving

where $V_{ai}^{\,l}$ and $V_{ai}^{\,m}$ are the InSAR LOS velocity of the $i$th points in the $l$th and the $m$th ascending tracks, respectively; $V_{di}^{\,n}$ and $V_{di}^{\,p}$ are the

$$
\begin{bmatrix} V_{ai}^{\,l} \\ \vdots \\ V_{ai}^{\,m} \\ \vdots \\ V_{di}^{\,n} \\ \vdots \\ V_{di}^{\,p} \\ \vdots \\ V_{ei} \\ \vdots \\ V_{ni} \\ \vdots \\ V_{vi} \\ \vdots \end{bmatrix}
=
\begin{bmatrix}
-\sin\theta_i^l\cos\phi^l & \dots & \Delta_i^l & \dots & 0 & \dots & 0 & \dots & x_i & y_i & 1 & 0 & 0 & 0 & 0 & 0 & 0 & 0 & 0 & 0 & \dots \\
& \vdots & & & & & & & & & & & & & & & & & & & \\
-\sin\theta_i^m\cos\phi^m & \dots & \Delta_i^m & \dots & 0 & \dots & 0 & \dots & 0 & 0 & 0 & x_i & y_i & 1 & 0 & 0 & 0 & 0 & 0 & 0 & \dots \\
& \vdots & & & & & & & & & & & & & & & & & & & \\
-\sin\theta_i^n\cos\phi^n & \dots & \Delta_i^n & \dots & 0 & \dots & 0 & \dots & 0 & 0 & 0 & 0 & 0 & 0 & x_i & y_i & 1 & 0 & 0 & 0 & \dots \\
& \vdots & & & & & & & & & & & & & & & & & & & \\
-\sin\theta_i^p\cos\phi^p & \dots & \Delta_i^p & \dots & 0 & \dots & 0 & \dots & 0 & 0 & 0 & 0 & 0 & 0 & 0 & 0 & 0 & x_i & y_i & 1 & \dots \\
& \vdots & & & & & & & & & & & & & & & & & & & \\
1 & \dots & 0 & \dots & 0 & \dots & 0 & \dots & 0 & 0 & 0 & 0 & 0 & 0 & 0 & 0 & 0 & 0 & 0 & 0 & \dots \\
& \vdots & & & & & & & & & & & & & & & & & & & \\
0 & \dots & 0 & \dots & 1 & \dots & 0 & \dots & 0 & 0 & 0 & 0 & 0 & 0 & 0 & 0 & 0 & 0 & 0 & 0 & \dots \\
& \vdots & & & & & & & & & & & & & & & & & & & \\
0 & \dots & 0 & \dots & 0 & \dots & 1 & \dots & 0 & 0 & 0 & 0 & 0 & 0 & 0 & 0 & 0 & 0 & 0 & 0 & \dots \\
& \vdots & & & & & & & & & & & & & & & & & & &
\end{bmatrix}
\begin{bmatrix} V_{ei} \\ \vdots \\ V_{\mathrm{subv}i} \\ \vdots \\ V_{ni} \\ \vdots \\ V_{vi} \\ \vdots \\ a^l \\ b^l \\ c^l \\ a^m \\ b^m \\ c^m \\ a^n \\ b^n \\ c^n \\ a^p \\ b^p \\ c^p \\ \vdots \end{bmatrix}
\quad (i,m,n,l,p \in \mathbb{N}) \qquad (3)
$$

InSAR LOS velocity of the $i$th points in the $n$th and the $p$th descending tracks, respectively; $V_{ei}$, $V_{ni}$, $V_{vi}$ and $V_{\text{sub}vi}$ are the east-west, the north-south, the vertical and the sub-vertical components of the GNSS velocity of the $i$th points, respectively; $V_{\hat{e}i}$, $V_{\hat{n}i}$, $V_{\hat{v}i}$ and $V_{\text{sub}\hat{v}i}$ are the modelled east-west, north-south, vertical and sub-vertical components of the $i$th points, respectively; $x_i$ and $y_i$ are the location of the $i$th points; $a$, $b$, $c$ are the modelled factors to determine the linear plane for each track.

In our specific case, we uniformly sample the InSAR points in the overlapping regions across tracks with a spacing of ~10 km, and obtain 632 points that have two ascending and one descending measurements, and 840 points that have two descending and one ascending measurements. We also incorporate 84 available GNSS points with the horizontal components and 23 GNSS observations of both the horizontal and vertical components in the inversion (Supplementary Fig. 16). We invert the solution using least squares with equal weight to each data point. The results show a good fit between the model and the observations, with the root mean square error <0.6 mm/yr (Supplementary Fig. 16).

We then remove the linear planes determined from the data points from each track to mosaic the InSAR velocity field and transform the merged LOS velocities to the Tarim reference frame (Supplementary Fig. 17). The mosaicked LOS velocities are more consistent in all overlapping regions between adjacent tracks, with the average standard deviation of velocity differences dropping from 1.7 mm/yr to 1.4 mm/yr after velocity mosaicking (Supplementary Fig. 18).

## A modified deep-fault model

A deep-fault model describes a fault extending through the entire lithosphere as a discrete plane, with slip occurring continuously in the narrow zone, similar to that caused by an infinitely long screw dislocation in an elastic half-space at depth[59]. The modified deep-fault model for the profiles showing strain accumulated along the ATF can be expressed as

$$v_p = \begin{cases} \frac{2S\gamma}{\pi}\arctan\left(\frac{x+l}{d_1}\right) - C\left(\frac{1}{\pi}\arctan\left(\frac{x+l}{d_2}\right)\right. \\ \left. -\text{H}(x+l)\right) + \theta(x+l) + a, & \text{if } x+l \geq 0 \\ \frac{2S(1-\gamma)}{\pi}\arctan\left(\frac{x+l}{d_1}\right) - C\left(\frac{1}{\pi}\arctan\left(\frac{x+l}{d_2}\right)\right. \\ \left. -\text{H}(x+l)\right) + a, & \text{if } x+l < 0 \end{cases} \quad (4)$$

$$\text{where} \quad \text{H}(x+l) = \begin{cases} 1 & \text{if } x+l \geq 0 \\ 0 & \text{if } x+l < 0 \end{cases} \quad \text{and} \quad \gamma = \frac{R_b}{(R_p+R_b)}$$

where $v_p$ is the fault-parallel velocities, $S$ is the slip rate, $x$ is the perpendicular distance to the fault trace, $l$ is the horizontal shift between the fault trace and buried dislocation, $d_1$ is the locking depth, $C$ is the creep rate, $d_2$ is the creep depth, H($x$) is the Heaviside function, $\theta$ is solved to correct the rotation effect, $\gamma$ represents the ratio of the rigidity in the Tibetan Plateau, $R_p$, and the Tarim block $R_b$, and $a$ is a static offset.

We solve for additional slip rate $S_2$, locking depth $d_3$, and buried dislocation shift $l_2$ on secondary faults as

$$v_p = \begin{cases} \frac{2S\gamma}{\pi}\arctan\left(\frac{x+l}{d_1}\right) - C\left(\frac{1}{\pi}\arctan\left(\frac{x+l}{d_2}\right)\right. \\ \left. -\text{H}(x+l)\right) + \theta(x+l) + \frac{S_2}{\pi}\arctan\left(\frac{x+l_2}{d_3}\right) + a, & \text{if } x+l \geq 0 \\ \frac{2S(1-\gamma)}{\pi}\arctan\left(\frac{x+l}{d_1}\right) - C\left(\frac{1}{\pi}\arctan\left(\frac{x+l}{d_2}\right)\right. \\ \left. -\text{H}(x+l)\right) + \frac{S_2}{\pi}\arctan\left(\frac{x+l_2}{d_3}\right) + a, & \text{if } x+l < 0 \end{cases} \quad (5)$$

$$\text{where} \quad \text{H}(x+l) = \begin{cases} 1 & \text{if } x+l \geq 0 \\ 0 & \text{if } x+l < 0 \end{cases} \quad \text{and} \quad \gamma = \frac{R_b}{(R_p+R_b)}$$

We sampled a posterior probability density function (PDF) of each model parameter given observed data and prior information through a Bayesian Markov chain Monte Carlo (MCMC) scheme. The likelihood function is assumed multivariate Gaussian with the variance-covariance matrix for the data estimated by calculating a 1-D experimental semi-variogram for the non-deforming region and fitting an exponential covariance function[60]. In each iteration of the MCMC, the algorithm draws a random walk step from a boxcar prior PDF and then scales this step by an optimised maximum step (different for each parameter), defined by an automatic step size selection process[61], to ensure an appropriate acceptance/rejection ratio for all parameters. Prior constraints for model parameters are: $0 < S < 30$ mm/yr, $0 < d_1 < 40$ km, $-100 < l < 100$ km, $-100 < a < 100$ mm/yr, $0.5 < \gamma < 1$, $0 < C < 10$ mm/yr, $0 < d_2 < 40$ km, $0 < S_2 < 30$ mm/yr, $0 < d_3 < 10$ km, $-300 < l_2 < 0$ km, $0 < \theta < 0.03$. In addition, we constrain the creep depth, $d_2$, to be shallower than the locking depth, $d_1$, and enforce the horizontal shift of the buried dislocation for the additional strain localisation in the south, $l_2$, to be greater than the shift to the strain along the ATF, $l$. We compare the new likelihood in each iteration to the likelihood of the previous solution using the Metropolis-Hastings algorithm[62,63] to determine whether the new trial should be accepted or rejected. As the model parameter vector is updated during the MCMC iterations, the inversion scheme populates the posterior PDF. We run the model for 300,000 iterations to ensure convergence on the most probable model and to have a sufficiently sampled PDF. We extract the maximum a posteriori solution and also the 68% confidence limits for each parameter from the posterior PDF.

## Strain rate estimation

Assuming that the strain in the direction perpendicular to each profile is zero, we can calculate the peak strain rate at the surface above the fault, $\dot{\varepsilon}$, from the estimated slip rate and locking depth along the main strand for each profile by differentiating the deep-fault model as

$$\dot{\varepsilon} = \frac{S}{2\pi d_1} \quad (6)$$

The estimate of strain rate is the rate expected from slip rate alone, ignoring any creep.

## Logarithmic decay trend

$$d = a\log\left(1 + \frac{\triangle t}{T}\right) \quad (7)$$

where $d$ is the relative LOS displacement, $a$ and $T$ are characteristic constants, and $\triangle t$ is the time since the earthquake occurred.

## Distributed shear zone model

In contrast to the deep-fault model, which features stable sliding of a single narrow fault beneath a locked elastic lid, the distributed shear zone model has a distributed deformation at depth. The long-term motion of the lower layer deforms in a ductile manner, resulting in the total deformation being distributed linearly across a broader shear zone compared to the strain concentrated on the single primary fault[64]. This model integrates a distribution of screw dislocations within one finite shear zone, assuming constant strain rate, such that

$$V_p = \frac{S}{2\pi(c-b)}\left\{\left(\pi(c-b) + 2((x+l)-b)\right)\arctan\frac{((x+l)-b)}{d_1}\right.$$
$$+ 2(c-(x+l))\arctan\frac{((x+l)-c)}{d_1} + d_1\log_{10}(d_1^2 + ((x+l)-c)^2)$$
$$\left. -d_1\log_{10}(d_1^2 + ((x+l)-b)^2)\right\} + a \quad (8)$$
$$\text{where} \quad b = -c$$

where $b$ and $c$ are half of the shear zone width on each side of the solved buried dislocation. Prior constraints in the MCMC sampler for model parameters are: $0 < S < 30$ mm/yr, $-150 < l < 150$ km, $-100 < a < 100$ mm/yr and $0 < c < 200$ km assuming a uniform probability distribution over each range. The locking depth, $d_1$, is fixed at 15 km.

## Data availability

The Sentinel-1 data is available from the Sentinels Scientific Data Hub of the European Space Agency (https://scihub.copernicus.eu). The HRES-ECMWF pressure level products are provided by the European Centre for Medium-Range Weather Forecasts (ECMWF) (https://www.ecmwf.int/en/forecasts/datasets/set-i). The seismicity data is available from the USGS (https://www.usgs.gov/programs/earthquake-hazards/earthquakes). The SRTM data is available from the USGS. The InSAR LOS velocity data along the ATF have been publicly archived on Figshare at https://doi.org/10.6084/m9.figshare.26288890.v1. The data supporting the findings of this study are provided in the Supplementary Information.

## Code availability

A software package of the spatially varying scaling method for InSAR tropospheric correction is available at https://github.com/Lin1119/ASVS (https://doi.org/10.5281/zenodo.12740442). The scripts of InSAR velocity mosaicking over adjacent tracks can be accessed at https://github.com/Lin1119/InSAR_mosaicking (https://doi.org/10.5281/zenodo.12740901).

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

## Acknowledgements

We acknowledge Dr Austin J. Elliott at the Earthquake Science Centre of the USGS for sharing with us the high-resolution fault trace data along the main strand of the ATF (https://doi.org/10.1016/j.tecto.2018.01.004). We thank Professor Gregory A. Houseman for providing valuable comments and suggestions on this work. We thank the support from the NERC through the Looking into the Continents from Space (LiCS) large grant (NE/K010867/1 to T.J.W. and A.H.). This work was also supported by the Centre for the Observation and Modelling of Earthquakes, Volcanoes and Tectonics (COMET) in the United Kingdom, a partnership between UK universities and the British Geological Survey. We acknowledge support from the Royal Society through a University Research Fellowship (UF150282 to J.R.E.). Some figures were generated using Generic Mapping Tools software (https://www.generic-mapping-tools.org).

## Author contributions

L.S. and A.H. conceived the research. L.S. processed the data, performed the modelling, and analysed the results. A.H. and L.S. designed the method of InSAR velocity mosaicking. A.H., J.R.E. and T.J.W. acquired the funding that supported this research and supervised the work. L.S. draughted the paper. All authors contributed to finalising and editing the paper.

## Competing interests

The authors declare no competing interests.
