## [Peer Review File · Nature Communications]

Variability in interseismic strain accumulation rate and style along the Altyn Tagh FaultREVIEWER COMMENTS

Reviewer #1 (Remarks to the Author):

The Altyn Tagh Fault (ATF) is an important fault in Asia and serves as the northern boundary of the Qinghai-Tibet Plateau. As early as 2000, Bendick et al. initiated geodetic surveys in this location. Subsequently, Shen et al. 2001; Zhang et al. 2007; He et al. 2013, and others conducted GNSS observations with enhanced precision. However, due to the harsh natural conditions, there has been a lack of observations on the deformation along the entire fault zone. The authors addressed this gap by utilizing InSAR technology. They presented the first InSAR-derived velocity field covering almost the entire length of the 1600 km-long Altyn Tagh Fault and analyzed the strain distribution. However, when compared to the extensive geodetic research conducted in the region, this study lacks sufficient novelty. Additionally, there are several significant issues within this paper that require revision.

There are three issues regarding the data processing. Firstly, as far as I know, LiCSAR processes InSAR data on each frame individually, which may result in the fault trace coinciding with the edge of a particular frame, making it challenging to remove the trend component. The authors mention that they merge frames within the same track to form a long track before generating interferograms using LiCSAR, to avoid possible inconsistent phase in the overlapping region. However, I did not observe this step in Supplementary Figure 7.

Secondly, concerning the baseline selection, the authors chose short temporal and spatial baselines for the time-series InSAR processing. However, for the Altyn Tagh Fault, which experiences displacements of several millimeters per year, using such short temporal baselines would result in displacement signals on each interferogram being less than a millimeter. On the other hand, noise phases like atmospheric delay phases could be in the range of several to tens of millimeters, significantly affecting the displacement-to-noise ratio of the data. Have the authors considered that the spatial baselines of Sentinel data are already short, and there is no need to restrict them, primarily focusing on the temporal baselines? For the northern boundary of the Qinghai-Tibet Plateau, which has a dry climate and almost no vegetation, selecting longer-term baselines, such as one or two years, can still yield well-coherent interferograms. The advantage of this approach is that it can significantly improve the displacement-to-noise ratio of the data by tenfold or more. Therefore, I believe that for the Altyn Tagh Fault, especially in the case of long-term tectonic deformation, longer temporal baselines hold more advantages over shorter ones.

Thirdly, during the InSAR processing, two of the most challenging aspects are atmospheric effects and orbital errors. The authors previously performed atmospheric correction for a section of the Altyn Tagh Fault (Shen et al. 2019). However, in this study, they mentioned using their previous method for removing atmospheric delay phases and showcased the RMS comparisons of interferograms in Supplementary Figure 9. It would be helpful to include the results of atmospheric correction for each track, before and after the correction, in the manuscript, to provide a more

visual demonstration of the effectiveness of this atmospheric correction method. Additionally, the authors mentioned applying linear corrections to the InSAR deformation field using GPS. However, in Figure 1(b), there are indications of subsidence (represented as negative values in blue) at both ends of the Altyn Tagh Fault. The authors attributed this to the south dipping Altyn Shan thrusts along the southern border of the Tarim Basin and hydrological processes or effects from the sand dunes. However, this explanation appears unsubstantiated.

Regarding the inversion of fault slip rates, the model proposed by Savage and Burford (1973) has been repeatedly used for the Altyn Tagh Fault, demonstrating its applicability in the region. However, for the segment of the Altyn Tagh Fault from 86° E to 90° E, due to the uplift of the Altyn Shan range, deformation becomes complex in this area, with displacement occurring not only along the Altyn Tagh Fault but also along the northern boundary faults of the Altyn Shan range and within the intermountain basins. Therefore, employing this simple inversion model may not be suitable for this particular segment, leading to unreliable inversion results.

Other minor issues include:

L31-34: Please include the published GNSS results, such as Shen Z K, Wang M, Li Y, et al. 2001; He J, Chéry J. 2008; Ge W P, Shen Z K, Molnar P, et al. 2022; Li Y, Nocquet J M, Shan X. 2022.

L189-191: The statement "The ATF intersects different terranes as well as distinctive geological structures such as sedimentary basins [37]" is too general. Have the authors taken into consideration that different segments of the Altyn Tagh Fault (ATF) may have formed under varying tectonic backgrounds? Providing more specific information would enhance the clarity of the statement.

L200-201: Please include this step for merging frames in Supplementary Fig. 7.

L217-223: Why did the author use the $a \cdot \sin(2\pi/365.25)t$ and $b \cdot \cos(2\pi/365.25)t$ to fit the annual seasonal freezing and thawing of the upper layer of permafrost in the Tibetan Plateau? Why not utilize the new map of permafrost distribution on the Tibetan Plateau provided by <https://doi.org/10.5194/tc-11-2527-2017-supplement?>

L252-254: Is the "anomalous" subsidence at the eastern and western ends of the Altyn Tagh Fault (ATF) in Figure 1(b) related to your transformation of GPS velocities from a Eurasia-fixed reference frame to a Tarim reference?

Reviewer #2 (Remarks to the Author):

This manuscript presents analysis of interseismic deformation due to the Altyn Tagh Fault (ATF) at the northern margin of the Tibetan Plateau. The authors use SAR data from the Sentinel-1 mission to

measure interseismic velocities along most of the ATF, and estimate its "geodetic" slip rate. They find slip rates slightly in excess of 10 mm/yr, much lower than the previously suggested "geologic" slip rates (~30 mm/yr), but comparable to slip rates previously reported based on the GNSS data. A relatively low slip rate of the ATF indicates that the Tibetan Plateau is undergoing significant internal deformation. Results in the manuscript represent a substantial amount of work, and should be of interest to a broad community, even though they mostly confirm the previously published GNSS data. The data analysis is sound, and conclusions are supported by the presented evidence. I have several technical comments, as detailed below.

1) lines 2-3. "Major intracontinental strike-slip faults that have developed adjacent to strong regions are usually considered as boundaries that mark discontinuities in the velocity field" - it's not clear what velocity field is meant here. The interseismic velocity field is not discontinuous (except in case of creeping faults). Also, not all major strike-slip faults occur at the boundaries of "strong regions". Please clarify.

2) lines 11-14. "Major earthquakes" are those with magnitude greater than 7. What is the significance of M7.6? The ATF may or may not be capable of producing supershear ruptures, but it's unclear how the supershear ruptures are related to the absence of earthquakes on the ATF in the last 120+ years. Please explain, or remove.

3) lines 26-27. Replace "from" with "based on" or "using". I.e., "... modelling based on/using ... measurements".

4) lines 62-64. Why using the "east-west" velocity component, as opposed to e.g. the fault-parallel velocity? The latter would avoid unnatural assumptions about the "north-south" component, as well as potential trade offs with the vertical component, e.g., Lindsey et al., Localized and distributed creep along the southern San Andreas Fault, *J. Geophys. Res.*, 119, 7909-7922, 2014. The fault strike may not be exactly constant, but using an average strike is likely to introduce smaller errors compared to neglecting the "north-south" component.

5) line 81. Please cite evidence for a decreased rigidity on the Tibet side (if there is any).

6) line 97. "a wider zone of strain localisation" sounds a bit odd. Consider: "a wider zone in which the interseismic strain is accommodated due to splitting of the fault into three strands".

7) lines 141, 328-220. Equation (7) is sometimes used to fit an early postseismic transient, but in general it's non-physical, as it has no "fully relaxed" limit. Note that the pre-multiplying factor a in eq. 7 is NOT the amplitude of the transient. Also, the relaxation time T should not be referred to as the Maxwell relaxation time, as the latter is specific to linear viscoelastic solids. For an "afterslip" time function, see e.g. eq. (17) in Barbot et al., Postseismic deformation due to the Mw6.0 2004 Parkfield earthquake: Stress-driven creep on a fault with spatially variable rate-and-state friction parameters, *J. Geophys Res.*, 114, B07405, 2009.

8) In the Discussion section, the authors mention variations in lithology and temperature as

potential controls on the effective width of shear zones. Presumably they mean the degree of strain localization below the brittle-ductile transition. A broader zone of interseismic deformation (as observed at the Earth's surface) can be interpreted in different ways, including variations in the degree of strain localization below the brittle-ductile transition, viscoelastic effects, and multiple sub-parallel fault strands. Is there any evidence that the observed variations in the width of the observed deformation zone along the ATF correlate with either local geology, or geotherm? If so, please mention. Note that in case of multiple fault strands, each fault can still have highly localized "roots" at depth, as suggested for some major faults in the Tibet interior, see e.g. Jin et al., Transient deformation excited by the 2021 M7.4 Maduo (China) earthquake: Evidence of a deep shear zone, *J. Geophys. Res.*, e2023JB026643, 2023. A reader would benefit from a more substantive discussion on possible causes of variations in the effective width of the interseismic deformation zone along the ATF, which is one of the main points of this manuscript.

9) line 323. The authors are not inverting for the strain rate; consider "Strain rate estimation" instead of "inversion".

Reviewer #3 (Remarks to the Author):

Shen et al 2023 Nat Comms

In this article, Shen et al propose a map of surface deformation rates over one of the longest tectonic fault in Tibet, the Altyn Tagh Fault (ATF). The discussion about the history of the construction of the Tibetan plateau and its rheology is still not closed and the ATF is one of the main elements subject to debate. In particular, one of the main question is whether deformation focuses near large lithospheric faults or rather distributes broadly, with repercussions on the general rheology of the crust in this region. The authors propose a large map of surface deformation rates over the ATF in order to evaluate the distribution of strain and address the question of strain focusing. They explore two different models, one that focuses strain on the fault trace and another that includes a broad, distributed deformation zone. Their conclusion, which is quite new in my opinion, is that the signature of the mechanical processes acting at depth vary along strike with fault sections where deformation extends over broad, 100-km-wide zone, while strain focuses right on the fault trace along other sections.

Although I think this piece of data is a very important element in the discussion, I find the paper not very clear and could benefit from some important adjustments in order to better illustrate the conclusions of the authors. In particular, it is quite difficult to see what features in the data are justifying one model or the other, hence the final conclusions. Furthermore, the organisation of the paper is not completely clear as some elements should be passed from the methods section to the main text to improve the flow of the paper. Finally, I would say that the main conclusion ("the role of the ATF as a main controlling fault ... is smaller than previously thought...") is too vague and should

be precised.

In conclusion, I think the main conclusion of the paper (i.e. variable style of deformation focusing along the ATF) is supported by the data although the figures and discussion could be improved and I believe this conclusion is an important contribution to the debate. Some elements should be clarified and figures improved but I think that, in overall, this paper only requires minor to moderate revisions.

Main comments:

- Although I personally haven't taken a side in the ongoing discussions about the rheology of Tibet, I have the impression that a lot of the disagreements stem from vague, incomplete and sometimes too rough statements that are not necessarily useful (forgive me the bluntness of this assertion). The main conclusion here is that the style of deformation along the ATF changes along strike. The corollary about the "role" of the ATF in the main text is actually quite vague and should either be clarified or simply removed.

- Furthermore, the discussion paragraph should be put to a better use. Currently, it reads as a list of potential deformation mechanisms that may explain shear zone width in general, but there is no mention of scales involved (can you explain 100 km-wide shear zones by varying the content phyllosilicates? That sounds unlikely) and there is no direct link with the current matter, i.e. the ATF. Do we see any potential geological variability along strike that would match with the variations in shear zone width? Is there a geographical link with the different suture zones? Could the broad region to the south west be related to the fact that several earthquakes in the area could have impacted strain rates locally through some long lasting post-seismic effect even if no relaxation is visible in the time series? Could the rheology of the Qaidam explain the broad region to the north east? Could the presence of the Altyn Shan explain the central broad deformation zone? Refactoring this discussion paragraph would allow to be a bit more precise in the discussion about the interactions between faults and diffuse deformation in the history of the Tibetan plateau.

- My main concern is about the presentation of the data and the model vs data fits. All the figures showing profiles are extremely small and readers cannot assess the validity of the models. I would suggest to have a few, large profiles with the corresponding fits, comparing the diffuse deformation to the dislocation models in the main text so that readers can build their own opinion. It would also be good to have a sense of density of points in the profiles. Currently, each point has the same "visual weight" in the plots and it look like the scatter is very large, while they must cluster around a more interpretable trend. Furthermore, since one of the objectives is to figure out whether deformation is distributed or not, you should also evaluate the possibility that it distributes over multiple faults (see Fialko 2006 or Dalaison et al 2023 for instance). Given the size of the figures, it is not possible to assess whether some additional gradients could be found in these broad deformation regions.

- Modeling assumptions should be better introduced in the main text, not only in the methods section, otherwise readers might be lost. Please describe simply the models in the main text, referring to equations but describing the role of the main parameters (slip rate, rigidity ratio, locking depth, ...) and how do they translate in the data space.

- Some clarification is required in the data used for the modeling. The Altyn Tagh fault is not purely

East-West and it reads as if you are using the EW velocities to fit the data in both modeling cases. If so, this is incorrect given the equations presented in the methods section (for instance, eq. 4 is for fault parallel velocities and should not be used on EW velocities). I would encourage you to actually use the LOS data directly in order to correctly propagate errors.

- The modeling exploration is about comparing two end-member types of model. The answer, as most of the time, lies in between the two since some large features of the strain rate cannot be explained by the shear zone model alone and vice versa. Therefore, why not trying a model in which you impose some of the faults (ATF, Altyn Shan thrust, ...) on top of a diffuse shear zone to see how these families of models tradeoff? This would allow to avoid unrealistic locking depths of more than 40 km or very large misfits for the strain rates related to the ATF (see strain rate excursions in Fig4).

Additional comments:

- Line 16: Some earlier attempts were made with InSAR (Jolivet et al 2008) but also GNSS data (He et al 2013)

- Line 48 starting from “which is the first time...”: The end of the sentence is not really relevant and probably wrong (see Lemrabet et al 2023 for instance).

- Line 61: Resampling on a 1x1 km is convenient but probably overrides all the details related to shallow aseismic slip and other potential strain focusing. How much does this downsampling affect the results?

- Line 63: Why east-west and not fault parallel?

- Line 76 and 78: Unclear which data is modeled? Fault parallel or east-west?

- Line 81: Add reference 47 as well.

- Fig 3: Caption is incomplete. The additional strain corresponds to other faults but how are they justified? Do these correspond to anything in the geology? What is the depth of creep rate? Since creep rate seems high in the region where locking depth is unrealistically large, isn't creep used to balance the fact that the model does not focus strain enough to fit the data? Therefore, is the main conclusion still supported?

- Line 106: There is no strain rate inversion in the paper. You describe surface strain rate as deriving directly from the locking depth and loading rate, not coming from an inversion. How do you include the effect of surface creep in that strain rate estimate (strain rate should be infinite at the fault if creep reaches the surface).

- Fig 4: At longitude 89, shear zone width is relatively small (how small?). However, the inferred locking depth is of ~40 km at the same location, which should look like a wide shear zone. There is a contradiction here. Please enlarge at least one of the profiles to show the details and illustrate whether you might prefer one model or the other, or both.

- Line 173-175: That's the vague statement I was referring to. There is an equivalent one in the abstract

- Line 192: Some conclusion would be helpful for readers

- Line 311: Which PDF are you sampling? What proposal are you using? Is that an adaptive MCMC? Are uncertainties on the InSAR used somewhere? A bit more details about the implementation is required.

- Line 341: “To better allow” (remove the s)

- Figure S4 and figure 4 of the paper: In a larger plot, it would be nice to see multiple realizations of the prediction instead of simply the MAP (or mean).

- Figure S5: Insets should be much larger. Currently, labels overlap each other. In addition, I can see multimodal marginals in the Horizontal shift parameter. Could this mean that several faults could be detected in the profiles? What do these various locations correspond to in the data? In such case, which value do you take (the MAP, the mean or one of the modes)?

Looking forward to read an improved version of this interesting manuscript. While I consider my review, I realize that I hereby provide a lot of comments. I believe these suggested improvements correspond to moderate revisions of the manuscript and the idea that strain focusing along active faults varies along strike is an interesting one, which deserves to be published in Nat Comms, in my humble opinion.

Cheers,

Romain Jolivet, PhD

École normale supérieure

REVIEWER COMMENTS

Reviewer #1 (Remarks to the Author):

The Altyn Tagh Fault (ATF) is an important fault in Asia and serves as the northern boundary of the Qinghai-Tibet Plateau. As early as 2000, Bendick et al. initiated geodetic surveys in this location. Subsequently, Shen et al. 2001; Zhang et al. 2007; He et al. 2013, and others conducted GNSS observations with enhanced precision. However, due to the harsh natural conditions, there has been a lack of observations on the deformation along the entire fault zone. The authors addressed this gap by utilizing InSAR technology. They presented the first InSAR-derived velocity field covering almost the entire length of the 1600 km-long Altyn Tagh Fault and analyzed the strain distribution. However, when compared to the extensive geodetic research conducted in the region, this study lacks sufficient novelty. Additionally, there are several significant issues within this paper that require revision.

There are three issues regarding the data processing. Firstly, as far as I know, LiCSAR processes InSAR data on each frame individually, which may result in the fault trace coinciding with the edge of a particular frame, making it challenging to remove the trend component. The authors mention that they merge frames within the same track to form a long track before generating interferograms using LiCSAR, to avoid possible inconsistent phase in the overlapping region. However, I did not observe this step in Supplementary Figure 7.

Response: We confirm that in our analysis frames are merged before generating interferograms. We revised Supplementary Figure 10 (formerly Supplementary Figure 7) to include the step of merging along-track frames. We also revised Supplementary Figure 1 to show the merged data coverage of each track.

Secondly, concerning the baseline selection, the authors chose short temporal and spatial baselines for the time-series InSAR processing. However, for the Altyn Tagh Fault, which experiences displacements of several millimeters per year, using such short temporal baselines would result in displacement signals on each interferogram being less than a millimeter. On the other hand, noise phases like atmospheric delay phases could be in the range of several to tens of millimeters, significantly affecting the displacement-to-noise ratio of the data. Have the authors considered that the spatial baselines of Sentinel data are already short, and there is no need to restrict them, primarily focusing on the temporal baselines? For the northern boundary of the Qinghai-Tibet Plateau, which has a dry climate and almost no vegetation, selecting longer-term baselines, such as one or two years, can still yield well-coherent interferograms. The advantage of this approach is that it can significantly improve the displacement-to-noise ratio of the data by tenfold or more. Therefore, I believe that for the Altyn Tagh Fault, especially in the case of long-term tectonic deformation, longer temporal baselines hold more advantages over shorter ones.

Response: The inversion approach using short temporal baseline interferogram does actually generate interferograms with long temporal baselines. As the accumulated displacement is an aggregation of short temporal data, it provides both short-term and long-term signals. We have included Supplementary Figure 13 to illustrate an example of the derived accumulated

displacement of descending track 92, which contains both short-term and long-term accumulated deformation signals. We have described these points in lines 338 to 341.

Thirdly, during the InSAR processing, two of the most challenging aspects are atmospheric effects and orbital errors. The authors previously performed atmospheric correction for a section of the Altyn Tagh Fault (Shen et al.2019). However, in this study, they mentioned using their previous method for removing atmospheric delay phases and showcased the RMS comparisons of interferograms in Supplementary Figure 9. It would be helpful to include the results of atmospheric correction for each track, before and after the correction, in the manuscript, to provide a more visual demonstration of the effectiveness of this atmospheric correction method.

Response: We have added plots in Supplementary Figure 12 to demonstrate the effect of atmospheric correction on individual epochs of each track, which are strongly influenced by tropospheric delays. The InSAR phase delay of these epochs is highly correlated with the topography before correction, indicating the presence of strong tropospheric delays. However, after removing the estimated tropospheric delay from the InSAR data, the phase no longer has strong correlations with the topography. We have described these points in lines 330 to 333.

Additionally, the authors mentioned applying linear corrections to the InSAR deformation field using GPS. However, in Figure 1(b), there are indications of subsidence (represented as negative values in blue) at both ends of the Altyn Tagh Fault. The authors attributed this to the south dipping Altyn Shan thrusts along the southern border of the Tarim Basin and hydrological processes or effects from the sand dunes. However, this explanation appears unsubstantiated.

Response: The reviewer may be referring to Figure 2b. Figure 2b shows the "sub-vertical" velocities that include contributions from both north-south and vertical movements. Negative values indicate either southward motion or subsidence. In the region north of the Altyn Tagh Fault from 81.5°E to 82.5°E, the observed sub-vertical motion could potentially be attributed to southward movement resulting from left-lateral strike-slip of the fault. The sub-vertical component observed over the eastern edge of the Altyn Tagh Fault does not appear to be tectonic. We interpret this as being associated with hydrological processes or effects from the sand dunes there. We have clarified these points in lines 86 to 91.

Regarding the inversion of fault slip rates, the model proposed by Savage and Burford (1973) has been repeatedly used for the Altyn Tagh Fault, demonstrating its applicability in the region. However, for the segment of the Altyn Tagh Fault from 86° E to 90° E, due to the uplift of the Altyn Shan range, deformation becomes complex in this area, with displacement occurring not only along the Altyn Tagh Fault but also along the northern boundary faults of the Altyn Shan range and within the intermountain basins. Therefore, employing this simple inversion model may not be suitable for this particular segment, leading to unreliable inversion results.

Response: We thank the reviewer for this comment. It is indeed necessary to consider the variable characteristics of the Altyn Tagh Fault in the modelling. The fault-parallel velocity profiles derived in Supplementary Figure 4 reveal additional strain localisations distributed over southern strands near the western portion of the ATF from 84°E to 85.5°E and the eastern

portion from 91°E to 92°E. Therefore, we applied the modified elastic half-space model (Equation 5) to these profiles, capable of solving for additional slip rate, locking depth, and buried dislocation shift on secondary faults. We have included these points in lines 103 to 106 and lines 119 to 121.

From 86°E to 90°E along the fault, we observe higher estimates of locking depth derived from the elastic half-space model, indicating a wider zone where interseismic strain is accommodated due to the splitting of the fault into three strands. Therefore, we applied a shear zone model (Equation 8) to capture the broader distributed deformation at depth. The modelling results reveal a wider shear zone of approximately 108 km between 87.5°E to 88.5°E in the location where the fault breaks into three parallel strands, and the wider shear zones explain the high estimates of the locking depth solved from the elastic half-space model in the region. We have clarified these points in line 203 and lines 215 to 218.

Other minor issues include:

L31-34: Please include the published GNSS results, such as Shen Z K, Wang M, Li Y, et al. 2001; He J, Chéry J. 2008; Ge W P, Shen Z K, Molnar P, et al. 2022; Li Y, Nocquet J M, Shan X. 2022.

Response: We have included these references in line 40.

L189-191: The statement "The ATF intersects different terranes as well as distinctive geological structures such as sedimentary basins [37]" is too general. Have the authors taken into consideration that different segments of the Altyn Tagh Fault (ATF) may have formed under varying tectonic backgrounds? Providing more specific information would enhance the clarity of the statement.

Response: We have added more details in the discussion section (lines 268 to 277) regarding the effects of geological variation on the shear zone behavior observed along different segments of the ATF.

L200-201: Please include this step for merging frames in Supplementary Fig. 7.

Response: We have included the step of merging along-track frames in the revised Supplementary Figure 10 (formerly Supplementary Figure 7).

L217-223: Why did the author use the $a \cdot \sin(2\pi/365.25)t$ and $b \cdot \cos(2\pi/365.25)t$ to fit the annual seasonal freezing and thawing of the upper layer of permafrost in the Tibetan Plateau? Why not utilize the new map of permafrost distribution on the Tibetan Plateau provided by <https://doi.org/10.5194/tc-11-2527-2017-supplement>?

Response: While the map would tell us where the permafrost is, the inclusion of the sinusoidal and cosinusoidal functions allows us to model the displacement caused by the freeze-thaw cycle and therefore better estimate the annual ground velocity. We have clarified this in lines 348 to 349.

L252-254: *Is the "anomalous" subsidence at the eastern and western ends of the Altyn Tagh Fault (ATF) in Figure 1(b) related to your transformation of GPS velocities from a Eurasia-fixed reference frame to a Tarim reference?*

Response: The transformation of GNSS measurement velocities from a Eurasia-fixed reference frame to a Tarim reference is applied to horizontal components. After the transformation, the GNSS horizontal estimates show clear left-lateral strike-slip motion along the eastern and western ends of the Altyn Tagh Fault (refer to Supplementary Figure 14). We have included these points in lines 377 to 381.

Figure 2b shows sub-vertical velocities that include contributions from both north-south and vertical movements. Negative values indicate either southward motion or subsidence. Therefore, the observed sub-vertical motion at the western end of the Altyn Tagh Fault could potentially be attributed to southward movement resulting from left-lateral strike-slip of the fault. The sub-vertical component observed over the eastern edge of the Altyn Tagh Fault does not appear to be tectonic. We interpret this as being associated with hydrological processes or effects from the sand dunes. We have clarified these points in lines 86 to 91.

Reviewer #2 (Remarks to the Author):

This manuscript presents analysis of interseismic deformation due to the Altyn Tagh Fault (ATF) at the northern margin of the Tibetan Plateau. The authors use SAR data from the Sentinel-1 mission to measure interseismic velocities along most of the ATF, and estimate its "geodetic" slip rate. They find slip rates slightly in excess of 10 mm/yr, much lower than the previously suggested "geologic" slip rates (~30 mm/yr), but comparable to slip rates previously reported based on the GNSS data. A relatively low slip rate of the ATF indicates that the Tibetan Plateau is undergoing significant internal deformation. Results in the manuscript represent a substantial amount of work, and should be of interest to a broad community, even though they mostly confirm the previously published GNSS data. The data analysis is sound, and conclusions are supported by the presented evidence. I have several technical comments, as detailed below.

1) lines 2-3. "Major intracontinental strike-slip faults that have developed adjacent to strong regions are usually considered as boundaries that mark discontinuities in the velocity field" - it's not clear what velocity field is meant here. The interseismic velocity field is not discontinuous (except in case of creeping faults). Also, not all major strike-slip faults occur at the boundaries of "strong regions". Please clarify.

Response: The major intracontinental strike-slip faults discussed here are those lying adjacent to relatively strong regions. The term "velocity field" refers to the distribution of long-term velocities between tectonic plates or blocks, with strike-slip faults playing a significant role in these interactions. We have clarified these points in lines 2 to 5.

2) lines 11-14. "Major earthquakes" are those with magnitude greater than 7. What is the significance of M7.6? The ATF may or may not be capable of producing supershear ruptures,

but it's unclear how the supershear ruptures are related to the absence of earthquakes on the ATF in the last 120+ years. Please explain, or remove.

Response: According to the United States Geological Survey (USGS) earthquake records, a pair of earthquakes occurred along the western portion of the ATF in 1924, with magnitudes of Mw 7.0 and Mw 7.2, respectively. In 1932, a Mw 7.6 earthquake occurred along the Changma fault, near the easternmost part of the ATF. Following the 1932 earthquake, no major earthquake (Mw > 7.0) has been recorded along the ATF. We have revised these points in lines 16 to 21. We have removed the section about "supershear".

3) lines 26-27. Replace "from" with "based on" or "using". I.e., "... modelling based on/using ... measurements".

Response: We have changed the text as suggested.

4) lines 62-64. Why using the "east-west" velocity component, as opposed to e.g. the fault-parallel velocity? The latter would avoid unnatural assumptions about the "north-south" component, as well as potential trade offs with the vertical component, e.g., Lindsey et al., Localized and distributed creep along the southern San Andreas Fault, J. Geophys. Res., 119, 7909-7922, 2014. The fault strike may not be exactly constant, but using an average strike is likely to introduce smaller errors compared to neglecting the "north-south" component.

Response: We use the plane defined by east-west and sub-vertical components to mosaic the tracks, allowing us to resolve the east-west velocities unambiguously. We have described this point in lines 75 to 79. We then assume that the east-west velocities are solely the result of fault-parallel motion. Based on this assumption, we estimate 1-D fault-parallel velocities profiles using a varying local strike perpendicular to the high-resolution fault trace of the ATF shown in Figure 1, derived from the estimated east-west velocity field. We choose to estimate fault-parallel velocities from only the east-west velocities because InSAR has low sensitivity to north-south motion, and with sparse GNSS we cannot unambiguously separate north-south velocities from vertical velocities. We have included these points in lines 93 to 99.

5) line 81. Please cite evidence for a decreased rigidity on the Tibet side (if there is any).

Response: We have referenced a study (Kao et al., 2001) in line 103, indicating that the Tarim Basin appears to have remained a rigid block, experiencing little or no shortening in comparison to the Tibet side. We have also included a reference (Jolivet et al., 2008) in line 103, which provides evidence of a rigidity decrease from the Tarim Basin north of the Altyn Tagh Fault to the Tibetan Plateau.

6) line 97. "a wider zone of strain localisation" sounds a bit odd. Consider: "a wider zone in which the interseismic strain is accommodated due to splitting of the fault into three strands".

Response: We have changed the text as suggested.

7) lines 141, 328-220. Equation (7) is sometimes used to fit an early postseismic transient, but in

general it's non-physical, as it has no "fully relaxed" limit. Note that the pre-multiplying factor a in eq. 7 is NOT the amplitude of the transient. Also, the relaxation time T should not be referred to as the Maxwell relaxation time, as the latter is specific to linear viscoelastic solids. For an "afterslip" time function, see e.g. eq. (17) in Barbot et al., Postseismic deformation due to the Mw6.0 2004 Parkfield earthquake: Stress-driven creep on a fault with spatially variable rate-and-state friction parameters, J. Geophys Res., 114, B07405, 2009.

Response: We thank the reviewer for these comments. We agree that 'a' and 'T' in Equation 7 represent two characteristic constants for each time series. We have revised these points in lines 470 to 471. While other equations are available to model time-dependent deformation, we find that the simple one we choose performs well. We have clarified this point in lines 189 to 191.

8) In the Discussion section, the authors mention variations in lithology and temperature as potential controls on the effective width of shear zones. Presumably they mean the degree of strain localization below the brittle-ductile transition. A broader zone of interseismic deformation (as observed at the Earth's surface) can be interpreted in different ways, including variations in the degree of strain localization below the brittle-ductile transition, viscoelastic effects, and multiple sub-parallel fault strands. Is there any evidence that the observed variations in the width of the observed deformation zone along the ATF correlate with either local geology, or geotherm? If so, please mention. Note that in case of multiple fault strands, each fault can still have highly localized "roots" at depth, as suggested for some major faults in the Tibet interior, see e.g. Jin et al., Transient deformation excited by the 2021 M7.4 Maduo (China) earthquake: Evidence of a deep shear zone, J. Geophys. Res., e2023JB026643, 2023. A reader would benefit from a more substantive discussion on possible causes of variations in the effective width of the interseismic deformation zone along the ATF, which is one of the main points of this manuscript.

Response: We have added more details in the discussion section regarding the effects of geological (lines 254 to 277) and geothermal variations (lines 278 to 285) on the shear zone behavior observed along the ATF.

9) line 323. The authors are not inverting for the strain rate; consider "Strain rate estimation" instead of "inversion".

Response: We have changed the text to "Strain rate estimation" as suggested.

Reviewer #3 (Remarks to the Author):

Shen et al 2023 Nat Comms

In this article, Shen et al propose a map of surface deformation rates over one of the longest tectonic fault in Tibet, the Altyn Tagh Fault (ATF). The discussion about the history of the construction of the Tibetan plateau and its rheology is still not closed and the ATF is one of the main elements subject to debate. In particular, one of the main question is whether deformation focuses near large lithospheric faults or rather distributes broadly, with repercussions on the general rheology of the crust in this region. The authors propose a large map of surface deformation rates over the ATF in order to evaluate the distribution of strain and address the

question of strain focusing. They explore two different models, one that focuses strain on the fault trace and another that includes a broad, distributed deformation zone. Their conclusion, which is quite new in my opinion, is that the signature of the mechanical processes acting at depth vary along strike with fault sections where deformation extends over broad, 100-km-wide zone, while strain focuses right on the fault trace along other sections.

Although I think this piece of data is a very important element in the discussion, I find the paper not very clear and could benefit from some important adjustments in order to better illustrate the conclusions of the authors. In particular, it is quite difficult to see what features in the data are justifying one model or the other, hence the final conclusions. Furthermore, the organisation of the paper is not completely clear as some elements should be passed from the methods section to the main text to improve the flow of the paper. Finally, I would say that the main conclusion (“the role of the ATF as a main controlling fault ... is smaller than previously thought...”) is too vague and should be precised.

In conclusion, I think the main conclusion of the paper (i.e. variable style of deformation focusing along the ATF) is supported by the data although the figures and discussion could be improved and I believe this conclusion is an important contribution to the debate. Some elements should be clarified and figures improved but I think that, in overall, this paper only requires minor to moderate revisions.

Main comments:

- Although I personally haven't taken a side in the ongoing discussions about the rheology of Tibet, I have the impression that a lot of the disagreements stem from vague, incomplete and sometimes too rough statements that are not necessarily useful (forgive me the bluntness of this assertion). The main conclusion here is that the style of deformation along the ATF changes along strike. The corollary about the “role” of the ATF in the main text is actually quite vague and should either be clarified or simply removed.

Response: We have removed this statement from the main text and abstract. Instead, we have added sentences quantifying the proportion of the length of the ATF that the shear zone width indicates accommodates all of the strain (lines 237 to 239). This highlights the important role of sub-parallel faults near the ATF in the overall deformation field (lines 240 to 244).

- Furthermore, the discussion paragraph should be put to a better use. Currently, it reads as a list of potential deformation mechanisms that may explain shear zone width in general, but there is no mention of scales involved (can you explain 100 km-wide shear zones by varying the content phyllosilicates? That sounds unlikely) and there is no direct link with the current matter, i.e. the ATF. Do we see any potential geological variability along strike that would match with the variations in shear zone width? Is there a geographical link with the different suture zones? Could the broad region to the south west be related to the fact that several earthquakes in the area could have impacted strain rates locally through some long lasting post-seismic effect even if no relaxation is visible in the time series? Could the rheology of the Qaidam explain the broad region to the north east? Could the presence of the Altyn Shan explain the central broad deformation zone? Refactoring this discussion paragraph would allow to be a bit more precise in

the discussion about the interactions between faults and diffuse deformation in the history of the Tibetan plateau.

Response: We have restructured the discussion section to address each of the questions suggested by the reviewer. Specifically, we discussed the effects of geological variability on the shear zone behaviour observed along the ATF (lines 254 to 260). We discussed how distinctive geological structures intersecting with the ATF can lead to rheological variations that influence fault localisation, including the presence of Altyn Shan (lines 268 to 274) and the rheology of the Qaidam Basin (lines 274 to 277). Additionally, we discussed the influence of higher heat flow on the wide shear zone behaviour (lines 278 to 285), such as its impact on the western part of the Qaidam Basin (lines 283 to 285). Furthermore, we explored the effect of past earthquake events on the fault and shear zone behaviour (lines 286 to 290). We have removed the discussion about phyllosilicates.

- My main concern is about the presentation of the data and the model vs data fits. All the figures showing profiles are extremely small and readers cannot assess the validity of the models. I would suggest to have a few, large profiles with the corresponding fits, comparing the diffuse deformation to the dislocation models in the main text so that readers can build their own opinion. It would also be good to have a sense of density of points in the profiles. Currently, each point has the same “visual weight” in the plots and it look like the scatter is very large, while they must cluster around a more interpretable trend. Furthermore, since one of the objectives is to figure out whether deformation is distributed or not, you should also evaluate the possibility that it distributes over multiple faults (see Fialko 2006 or Dalaison et al 2023 for instance). Given the size of the figures, it is not possible to assess whether some additional gradients could be found in these broad deformation regions.

Response: We have added plots in Figure 4 and Supplementary Figure 5 showing three enlarged profiles for each broad shear zone, at 84°E, 87.5°E and 91°E, respectively. In these enlarged profiles, we now show the mean value and the 95% confidence range of the data profile to provide a sense of density of data points. Additionally, we have included the position of mapped geological structures intersecting with the profile, as well as the estimated buried dislocation of the ATF to allow assessment of whether strain concentrates on other nearby sub-parallel faults. We have included these points in lines 204 to 211.

- Modeling assumptions should be better introduced in the main text, not only in the methods section, otherwise readers might be lost. Please describe simply the models in the main text, referring to equations but describing the role of the main parameters (slip rate, rigidity ratio, locking depth, ...) and how do they translate in the data space.

Response: We have included descriptions about the models in the main text at lines 111 to 121.

- Some clarification is required in the data used for the modeling. The Altyn Tagh fault is not purely East-West and it reads as if you are using the EW velocities to fit the data in both modeling cases. If so, this is incorrect given the equations presented in the methods section (for

instance, eq. 4 is for fault parallel velocities and should not be used on EW velocities). I would encourage you to actually use the LOS data directly in order to correctly propagate errors.

Response: We used the fault-parallel velocities converted from the east-west velocity field in both modelling cases, which we have clarified in lines 93 to 96 and lines 199 to 201. We choose to estimate fault-parallel velocities from only the east-west velocities because InSAR has low sensitivity to north-south motion, and with sparse GNSS we cannot unambiguously separate north-south velocities from vertical velocities. It is a good idea to use the LOS data directly, and we will include it into our future modelling work.

- The modeling exploration is about comparing two end-member types of model. The answer, as most of the time, lies in between the two since some large features of the strain rate cannot be explained by the shear zone model alone and vice versa. Therefore, why not trying a model in which you impose some of the faults (ATF, Altyn Shan thrust, ...) on top of a diffuse shear zone to see how these families of models tradeoff? This would allow to avoid unrealistic locking depths of more than 40 km or very large misfits for the strain rates related to the ATF (see strain rate excursions in Fig4).

Response: Thank you. The current estimated strain rates for each profile shown in Figure 4b have indicated multi-peak strain rates over the three broad shear zones (lines 226 to 228), suggesting that the estimation can capture both the peak strain on the main strand of the ATF and wide, diffused strain over the broad shear zone. Therefore, we consider it a good direction for future work to impose some of the faults on top of a diffuse shear zone in the modelling.

Additional comments:

- Line 16: Some earlier attempts were made with InSAR (Jolivet et al 2008) but also GNSS data (He et al 2013)

Response: We have added these references (line 35).

- Line 48 starting from "which is the first time...": The end of the sentence is not really relevant and probably wrong (see Lemrabet et al 2023 for instance).

Response: The published InSAR research over the Altyn Tagh Fault usually addresses only a portion of the fault rather than its entire length. However, we have changed the wording to focus on the resolution provided by InSAR. We have now state (lines 56 to 58), "which is the first time a large-scale analysis covering almost the entire length of the fault has been carried out with a resolution sufficient to identify areas of strain localisation."

- Line 61: Resampling on a 1x1 km is convenient but probably overrides all the details related to shallow aseismic slip and other potential strain focusing. How much does this downsampling affect the results?

Response: We have added Supplementary Figure 3 to show a comparison of LOS velocities before and after resampling in ascending and descending directions, respectively. We calculated

velocity differences along short fault-parallel velocity profiles with a length of 10 km on either side of the ATF, and the results indicate a negligible difference after resampling. We have added this point in lines 72 to 75.

- *Line 63: Why east-west and not fault parallel?*

Response: Please see response to Reviewer #2 Point 4.

- *Line 76 and 78: Unclear which data is modeled? Fault parallel or east-west?*

Response: We modelled the fault-parallel velocities, which we have clarified in lines 109 to 111 and lines 199 to 201.

- *Line 81: Add reference 47 as well.*

Response: We have included the reference as suggested at line 103.

- *Fig 3: Caption is incomplete. The additional strain corresponds to other faults but how are they justified? Do these correspond to anything in the geology? What is the depth of creep rate? Since creep rate seems high in the region where locking depth is unrealistically large, isn't creep used to balance the fact that the model does not focus strain enough to fit the data? Therefore, is the main conclusion still supported?*

Response: The estimated additional buried dislocations for the profiles from 84°E to 85.5°E are located ~50 km to 150 km south of the ATF, where other sinistral faults are mapped (Taylor et al., 2009). For the profiles over the eastern portion from 91°E to 92°E, the buried dislocation shifts ~130 km southward to a region that features other multiple fault strands. We have clarified these points in the caption of Figure 3 and in lines 136 to 140.

We have included the estimated creep depth in Figure 3.

We have added a Supplementary Figure 7 indicates that the estimated creep rate is not highly correlated with the estimated locking depth, as the locking depth of areas with higher estimates of creep rate varies from 15 km to 40 km. We have also included this point in lines 146 to 150.

- *Line 106: There is no strain rate inversion in the paper. You describe surface strain rate as deriving directly from the locking depth and loading rate, not coming from an inversion. How do you include the effect of surface creep in that strain rate estimate (strain rate should be infinite at the fault if creep reaches the surface).*

Response: Our estimate of strain rate is the rate expected from slip rate alone, ignoring any creep. We have clarified this in the text (line 468).

- *Fig 4: At longitude 89, shear zone width is relatively small (how small?). However, the inferred locking depth is of ~40 km at the same location, which should look like a wide shear zone. There*

is a contradiction here. Please enlarge at least one of the profiles to show the details and illustrate whether you might prefer one model or the other, or both.

Response: The estimated shear zone width is ~0.6 km at 89°E and ~4.2 km at 89.5°E, which is much narrower than the two neighbouring sections with a broader shear zone width. The estimated narrow shear zone width is likely a result of the relatively noisy data in that area, which cannot distinguish between a possible shallow creep using the elastic half-space model or a localised single fault using the shear zone model. We have clarified these points in lines 219 to 225.

We have added plots in Figure 4 and Supplementary Figure 5 showing three enlarged profiles for each broad shear zone, located at 84°E, 87.5°E, and 91°E, respectively.

- Line 173-175: That's the vague statement I was referring to. There is an equivalent one in the abstract

Response: Please see our response to reviewer's first comment.

- Line 192: Some conclusion would be helpful for readers

Response: We have added a conclusion section in lines 291 to 309 to summarise our work as suggested.

- Line 311: Which PDF are you sampling? What proposal are you using? Is that an adaptive MCMC? Are uncertainties on the InSAR used somewhere? A bit more details about the implementation is required.

Response: We have now added these details in lines 437 to 462.

- Line 341: "To better allow" (remove the s)

Response: Done.

- Figure S4 and figure 4 of the paper: In a larger plot, it would be nice to see multiple realizations of the prediction instead of simply the MAP (or mean).

Response: We have included the 95% model confidence range into the plots of enlarged profiles (Figure 4 and Supplementary Figure 5), providing multiple realizations of the prediction.

- Figure S5: Insets should be much larger. Currently, labels overlap each other. In addition, I can see multimodal marginals in the Horizontal shift parameter. Could this mean that several faults could be detected in the profiles? What do these various locations correspond to in the data? In such case, which value do you take (the MAP, the mean or one of the modes)?

Response: We have corrected the overlapping labels in Supplementary Figure 6 (formerly Supplementary Figure 5). The horizontal shift estimated for the main strand of the ATF

represents the distance between the fault trace and buried dislocation. The estimated multimodal marginals shown in the posterior probability distributions suggest that the strain could be distributed along multiple strands of the ATF (e.g., at 85.5°E and 91°E) or may be influenced by relatively noisy data where there are no mapped multiple strands (e.g., at 81.5°E and 89°E). We have included these points in lines 140 to 143.

The estimated buried dislocation is located south of the fault from 81.5°E to 83°E. It then interweaves with the fault trace along its central portion before shifting to the north of the fault from 91.5°E eastward. We have included these points in lines 133 to 136.

In our modeling, we computed the Maximum A Posteriori (MAP) solution to characterise the localised strain of the fault. We have clarified this in lines 461 to 462.

Looking forward to read an improved version of this interesting manuscript. While I consider my review, I realize that I hereby provide a lot of comments. I believe these suggested improvements correspond to moderate revisions of the manuscript and the idea that strain focusing along active faults varies along strike is an interesting one, which deserves to be published in Nat Comms, in my humble opinion.

Cheers,

Romain Jolivet, PhD

École normale supérieure

Response: Many thanks for this helpful review.

REVIEWERS' COMMENTS

Reviewer #1 (Remarks to the Author):

The author has addressed the majority of my previous concerns, and the overall argumentation in the manuscript has been strengthened. However, there are a few remaining minor issues that I hope the author can address:

1. Abstract: Regarding the sentence "This contrasts to the North Anatolian and central San Andreas faults, where strain rate varies little along the entire length.", I think that this sentence may not be essential to the core focus of the paper. The author could consider removing it to make the abstract more concise and focused.

2. The author has compared the GNSS results and the InSAR results in the text (as shown in Supplementary Figure 4). I suggest that the author should include the locations of the GNSS stations used in the analysis in either Supplementary Figure 4 or Figure 1. This additional spatial information would greatly facilitate the reader's understanding of the spatial context of the data comparison.

3. L145-146: "Except for the 2 mm/yr creep rate observed at the westernmost end of the fault and in the region from 88.5°E to 90.5°E", I recommend the author provide a brief discussion to address the following two points, as this phenomenon is likely to be of significant interest to many readers:

- Confirm whether the 2 mm/yr creep rate at the westernmost end of the fault is related to the post-seismic deformation following the February 2014 Mw 6.9 Yutian earthquake.
- Explain the reason for the observed creep in the region from 88.5°E to 90.5°E, as the GNSS results reported by Ge et al. (2022, <https://doi.org/10.1029/2022JB024216>) do not show such a creep phenomenon.

Reviewer #2 (Remarks to the Author):

The authors have done a good job with their revision. I have a few comments on the revised manuscript that should be fairly easy to address.

1) Lines 43-44. "For instance, the geodetic measurements are generally 2–3 times less than some of the Quaternary measurements". Here, the comparison is between slip rates, not "measurements" (which are not directly comparable). Consider: "For instance, slip rates inferred from geodetic data are generally a factor of 2 to 3 lower than those inferred from geologic data".

2) Lines 111-113. "To explain the asymmetry in the interseismic velocities on each side of the fault, we incorporate an asymmetry coefficient in the model to characterise the different rigidity between the Plateau and the Tarim Basin." It's not clear from the text whether the "asymmetry coefficient" is

solved for, or assumed based on some available data (e.g., seismic velocities). If it's solved for (which I think is the case), please provide the estimated values of the rigidity ratio, and discuss how realistic they are.

3) Lines 216-218. "The wider shear zones explain the high estimates for the locking depth solved from the elastic half-space model in these areas" - it's not clear what shear zone model the authors are referring to here - several discrete faults (Line 421), or a "distributed" shear zone (Line 473). Partly, the confusion is due to terminology. The two models used by the authors are called "A modified elastic half-space model", and "a shear zone". In fact, both are models of a shear zone, and both assume elastic half-space. The main difference between these models is the degree of strain localization below the locking depth. To avoid confusion, the authors may wish to refer to these models as to discrete vs distributed shear zones. I suppose the rationale for using a distributed shear zone is that a discrete shear zone with several faults (based on geologically mapped fault traces) still produces locking depths that are too high? Please clarify.

4) Lines 228-230. "The results suggest that the strain over the wide shear zones could be distributed along multiple strands of the ATF or across other fault strands that are away from the ATF" - here, it would be instructive to mention examples of other fault systems that split into multiple strands, such as the northern Alpine or southern San Andreas, to point out that the ATF is not unique in this regard. E.g., at the end of the sentence: "... that are away from the ATF, similar to ...".

Reviewer #3 (Remarks to the Author):

In general, the authors have nicely addressed my comments and I believe this article is almost ready for publication. I find the use of a shear zone model over broad regions instead of the classic screw dislocation model quite nice and I would be very happy to see whether a 3D version of such model could be derived later (some kind of augmented block model with shear zones). However, two points remain incorrect in my opinion. These do not require much work but must be corrected, in my opinion.

The effect of the downsampling on the estimation of the shallow behavior of the fault is not clearly explained. In particular, I don't understand the test on resampling (Fig S7). How can one compute a velocity difference between non downsampled and downsampled without resampling? Since there is probably some resampling involved, the test just shows that the final velocity field is equivalent to the original one when they are both resampled. Maybe I did not understand what was done, but if this is the case, I don't get it.

The source of the data used for the modeling is, in my opinion, incorrect. InSAR is not "insensitive" to north-south motion. It is less sensitive but the actual zero plan strikes 13 to 15° to the east or west, depending on the LOS. Furthermore, the fault strikes at about 30° E so part of the signal, is

affecting the LOS measurements. One must therefore use the LOS measurement as data input for the modeling part rather than the fault-parallel-derive-from-the-east-west field. This easy fix, which involves a simple projection, will not change the shape of the shear zones but will affect slightly the slip rates. In particular, one can see that there is systematic differences between the fault parallel GNSS-derived displacements and the InSAR one (around 86° for instance, Fig S4) and I suspect this discrepancy comes from the assumption. Finally, asserting that NS motion cannot be measured with InSAR is wrong as it is possible to measure quite low slip rates on faults that show a relatively poor orientation when done carefully (check the papers on the Chaman fault, for instance, by Fattahi, Barnhart or Dalaison over the years with different constellations).

Sincerely,
Romain Jolivet
École normale supérieure

REVIEWERS' COMMENTS

Reviewer #1 (Remarks to the Author):

The author has addressed the majority of my previous concerns, and the overall argumentation in the manuscript has been strengthened. However, there are a few remaining minor issues that I hope the author can address:

1. Abstract: Regarding the sentence "This contrasts to the North Anatolian and central San Andreas faults, where strain rate varies little along the entire length.", I think that this sentence may not be essential to the core focus of the paper. The author could consider removing it to make the abstract more concise and focused.

Response: We have removed mention of the NAF and SAF and revised the text to: "We find that localisation of strain is actually variable, in contrast to other major strike-slip faults that show little variation, with strain concentrated at the fault for some sections and distributed over broad (>100 km) shear zones for others.", which makes the abstract more concise and focused.

2. The author has compared the GNSS results and the InSAR results in the text (as shown in Supplementary Figure 4). I suggest that the author should include the locations of the GNSS stations used in the analysis in either Supplementary Figure 4 or Figure 1. This additional spatial information would greatly facilitate the reader's understanding of the spatial context of the data comparison.

Response: We have added a plot of GNSS stations in Supplementary Figure 4 as suggested.

3.L145-146: "Except for the 2 mm/yr creep rate observed at the westernmost end of the fault and in the region from 88.5°E to 90.5°E", I recommend the author provide a brief discussion to address the following two points, as this phenomenon is likely to be of significant interest to many readers:

a. Confirm whether the 2 mm/yr creep rate at the westernmost end of the fault is related to the post-seismic deformation following the February 2014 Mw 6.9 Yutian earthquake.

Response: To investigate the impact of postseismic deformation following the February 2014 Mw 6.9 Yutian strike-slip earthquake, which occurred nine months before the InSAR observations of this study, we calculate the time series of relative LOS displacement between two sites located around 30 km apart, either side of the south-western segment. We calculate the average LOS displacement of points within 2 km distance to each site from the tropospheric corrected single primary interferograms, to form a time series of relative LOS displacement between the two sites, to which we fit a linear trend. Shown as the Supplementary Figure 9, we find a generally consistent rate in the ascending track. For the descending track, the displacement in the early time series, 673 days before the earthquake occurred, has a systematic bias,

indicating a higher rate before November 2015 compared to the later time series. We also fit a logarithmic decay trend to the time series. Compared to the linear model, however, there is no significant improvement when fitting the data with the logarithmic model. This suggests that the postseismic deformation is hard to distinguish from the long term interseismic deformation. Moreover, the estimated creep rate at the near-field region of 82°E is much lower, at 0.8 mm/yr. Therefore, the higher creep rate estimated at the westernmost end of 81.5°E is not likely caused by the impact of the postseismic deformation. We have included these points in lines 181 to 200.

b. Explain the reason for the observed creep in the region from 88.5°E to 90.5°E, as the GNSS results reported by Ge et al. (2022, <https://doi.org/10.1029/2022JB024216>) do not show such a creep phenomenon.

Response: Although shallow creep is not observed in the modelling results from the GNSS measurements between 88°E and 91°E, this could be due to the use of a wider profile (~300 km), which may be too smoothed to detect small-scale shallow creep. We have described these points in lines 142 to 145.

Reviewer #2 (Remarks to the Author):

The authors have done a good job with their revision. I have a few comments on the revised manuscript that should be fairly easy to address.

1) Lines 43-44. "For instance, the geodetic measurements are generally 2–3 times less than some of the Quaternary measurements". Here, the comparison is between slip rates, not "measurements" (which are not directly comparable). Consider: "For instance, slip rates inferred from geodetic data are generally a factor of 2 to 3 lower than those inferred from geologic data".

Response: We have revised the text to: "For instance, slip rates inferred from geodetic data are generally a factor of 2 to 3 lower than those inferred from geologic data," as suggested.

2) Lines 111-113. "To explain the asymmetry in the interseismic velocities on each side of the fault, we incorporate an asymmetry coefficient in the model to characterise the different rigidity between the Plateau and the Tarim Basin." It's not clear from the text whether the "asymmetry coefficient" is solved for, or assumed based on some available data (e.g., seismic velocities). If it's solved for (which I think is the case), please provide the estimated values of the rigidity ratio, and discuss how realistic they are.

Response: We confirm that we solve for the asymmetry coefficient in the model. To clarify, we have revised the text to "To explain the asymmetry in interseismic velocities on each side of the

fault, we solve for a rigidity ratio as an asymmetry coefficient in the model to characterise the differing rigidity between the Plateau and the Tarim Basin.”

We have included Supplementary Figure 8, showing the estimated values of the rigidity ratio along the Altyn Tagh Fault. As expected, we observe a reduction in rigidity from the Tarim Basin to the Tibetan Plateau, with an average rigidity ratio of 0.60. This agrees with the understanding that the Tarim Basin was relatively stable during the Cenozoic [34]. Moreover, we identify a higher rigidity ratio of up to 0.99 in the west-central segment from 83°E to 86°E, indicating a significant contrast in rigidity across this region. We have added these points in lines 149 to 154.

3) Lines 216-218. "The wider shear zones explain the high estimates for the locking depth solved from the elastic half-space model in these areas" - it's not clear what shear zone model the authors are referring to here - several discrete faults (Line 421), or a "distributed" shear zone (Line 473). Partly, the confusion is due to terminology. The two models used by the authors are called "A modified elastic half-space model", and "a shear zone". In fact, both are models of a shear zone, and both assume elastic half-space. The main difference between these models is the degree of strain localization below the locking depth. To avoid confusion, the authors may wish to refer to these models as to discrete vs distributed shear zones. I suppose the rationale for using a distributed shear zone is that a discrete shear zone with several faults (based on geologically mapped fault traces) still produces locking depths that are too high? Please clarify.

Response: The “modified elastic half-space model” describes a fault extending through the entire lithosphere as a discrete plane, with slip occurring continuously in the narrow zone, similar to that caused by an infinitely long screw dislocation in an elastic half-space at depth.

The “shear zone model” has deformation localised onto the discrete fault surface in the upper crust but distributed in wide shear zones at depth beneath the locked elastic lid. Therefore, the long-term motion of the upper crust is identical to that in the “modified elastic half-space model”, whereas the lower layer deforms in a ductile manner, resulting in the total deformation being distributed linearly.

To distinguish between them, we use the term “modified deep-fault model” to refer to the “modified elastic half-space model,” and the term “distributed shear zone model” to refer to the “shear zone model.”

The rationale for incorporating a distributed shear zone model is to investigate the distributed shear strain in a broader shear zone, as the results from the modified deep-fault model suggest that the strain along some portions of the ATF is likely widely distributed.

We have described these points in lines 202 to 205, lines 422 to 424, and lines 467 to 471 to explain them explicitly.

4) Lines 228-230. "The results suggest that the strain over the wide shear zones could be distributed along multiple strands of the ATF or across other fault strands that are away from the ATF" - here, it would be instructive to mention examples of other fault systems that split into multiple strands, such as the northern Alpine or southern San Andreas, to point out that the ATF is not unique in this regard. E.g., at the end of the sentence: "... that are away from the ATF, similar to ...".

Response: We have revised the text to "The results suggest that strain over wide shear zones could be distributed along multiple strands of the ATF or across other fault strands away from the ATF, where similar split strands can be observed in other fault systems such as the northern Alpine Fault [41] and the southern San Andreas Fault [42].", as suggested.

Reviewer #3 (Remarks to the Author):

In general, the authors have nicely addressed my comments and I believe this article is almost ready for publication. I find the use of a shear zone model over broad regions instead of the classic screw dislocation model quite nice and I would be very happy to see whether a 3D version of such model could be derived later (some kind of augmented block model with shear zones). However, two points remain incorrect in my opinion. These do not require much work but must be corrected, in my opinion.

The effect of the downsampling on the estimation of the shallow behavior of the fault is not clearly explained. In particular, I don't understand the test on resampling (Fig S7). How can one compute a velocity difference between non downsampled and downsampled without resampling? Since there is probably some resampling involved, the test just shows that the final velocity field is equivalent to the original one when they are both resampled. Maybe I did not understand what was done, but if this is the case, I don't get it.

Response: We compare the velocities before and after resampling along short fault-parallel velocity profiles with a length of 10 km on either side of the fault. To avoid any further resampling in the comparison, we show the velocity of points located within a spatial distance of 5 metres before and after resampling, individually, instead of the differences, in the new Supplementary Figure 3. The results also indicate a negligible difference in both ascending and descending directions before and after resampling. We have revised the text in lines 69 to 75 to clarify these points.

The source of the data used for the modeling is, in my opinion, incorrect. InSAR is not "insensitive" to north-south motion. It is less sensitive but the actual zero plan strikes 13 to 15° to the east or west, depending on the LOS. Furthermore, the fault strikes at about 30° E so part of the signal, is affecting the LOS measurements. One must therefore use the LOS measurement

as data input for the modeling part rather than the fault-parallel-derive-from-the-east-west field. This easy fix, which involves a simple projection, will not change the shape of the shear zones but will affect slightly the slip rates. In particular, one can see that there is systematic differences between the fault parallel GNSS-derived displacements and the InSAR one (around 86° for instance, Fig S4) and I suspect this discrepancy comes from the assumption. Finally, asserting that NS motion cannot be measured with InSAR is wrong as it is possible to measure quite low slip rates on faults that show a relatively poor orientation when done carefully (check the papers on the Chaman fault, for instance, by Fattahi, Barnhart or Dalaison over the years with different constellations).

Response: We have followed the reviewer's suggestion to perform the modelling using the fault-parallel velocities directly derived from the LOS measurements and revised the manuscript accordingly. As expected, the new results show consistent variation in slip rate and shear zone width with slightly higher slip rates. The RMSE of differences between the GNSS-derived fault-parallel velocities and the InSAR-derived fault-parallel velocities slightly decreased from 2.0 mm/yr to 1.9 mm/yr at 86°E, indicating a small impact of the previous assumption.

We have clarified that we use the InSAR LOS measurements as the source of modelling in lines 93 to 95. To avoid confusion, we removed the section about “InSAR having low sensitivity to north-south motion” in the main text.

*Sincerely,
Romain Jolivet
École normale supérieure*

Response: We thank all reviewers for the very constructive and helpful reviews.